# Social synchronization of brain activity increases during eye-contact

Caroline Di Bernardi Luft [1✉], Ioanna Zioga[1,2], Anastasios Giannopoulos [3], Gabriele Di Bona [4], Nicola Binetti[1], Andrea Civilini[4], Vito Latora [4,5,6,7] & Isabelle Mareschal[1]

Humans make eye-contact to extract information about other people's mental states, recruiting dedicated brain networks that process information about the self and others. Recent studies show that eye-contact increases the synchronization between two brains but do not consider its effects on activity within single brains. Here we investigate how eye-contact affects the frequency and direction of the synchronization within and between two brains and the corresponding network characteristics. We also evaluate the functional relevance of eye-contact networks by comparing inter- and intra-brain networks of friends vs. strangers and the direction of synchronization between leaders and followers. We show that eye-contact increases higher inter- and intra-brain synchronization in the gamma frequency band. Network analysis reveals that some brain areas serve as hubs linking within- and between-brain networks. During eye-contact, friends show higher inter-brain synchronization than strangers. Dyads with clear leader/follower roles demonstrate higher synchronization from leader to follower in the alpha frequency band. Importantly, eye-contact affects synchronization between brains more than within brains, demonstrating that eye-contact is an inherently social signal. Future work should elucidate the causal mechanisms behind eye-contact induced synchronization.

[1] School of Biological and Behavioural Sciences, Queen Mary, University of London, London E1 4NS, United Kingdom. [2] Donders Institute for Brain, Cognition and Behaviour, Radboud University, Nijmegen, The Netherlands. [3] School of Electrical and Computer Engineering, National Technical University of Athens (NTUA), Athens, Greece. [4] School of Mathematical Sciences, Queen Mary University of London, London E1 4NS, United Kingdom. [5] Dipartimento di Fisica ed Astronomia, Università di Catania and INFN, I-95123 Catania, Italy. [6] The Alan Turing Institute, The British Library, London NW1 2DB, United Kingdom. [7] Complexity Science Hub, Josefstäadter Strasse 39, A 1080 Vienna, Austria. ✉email: c.luft@qmul.ac.uk

Human and non-human primates' gaze is drawn to others' eyes[1,2]. While non-human primates have a pigmented sclera, human's sclera are white[3]. This morphological difference allows humans to extract a wealth of information from our conspecific's eyes, which may shape our social interactions. For instance, humans can detect eye contact from a longer distance than nonhuman primates[4] and use this information to infer other people's mental states and intentions (for a review see[5]). The brain regions involved in eye-contact overlap with structures in the social brain network[6], including the ventral and medial prefrontal cortex, superior temporal gyrus, fusiform gyrus, cingulate gyrus and amygdala (for a review see[7]), suggesting that mutual eye contact is key for inferring others' emotions and intentions. The perception of direct eye contact in humans is consistently found to involve the superior temporal sulcus (STS)[8–10], a region, which is a key part of the mentalising network that is involved in tasks that require making inferences about the mental states of others[11]. Research has made remarkable progress towards understanding how eye contact is processed in a single (perceiver's) brain, but eye contact is an interactive process between two people. More recently, we have begun to extend this understanding to multiple brains—for example, the synchronization of activity between two brains has been found to increase during eye contact[12–14]. However, we still do not know how both intra- and inter-brain activity is integrated, nor the functional role of this synchronised activity.

To address this, it is important to examine the activity of two brains simultaneously, through a process known as Hyperscanning[15–19]. A classical Hyperscanning EEG study demonstrated that the brains of two people interacting in an imitation paradigm synchronize in a few frequencies, including alpha mu rhythms, beta, and gamma. Hyperscanning studies have shown that higher synchronization between brains (e.g., inter-brain activity) is associated with more effective social interactions[20–27]. For example, higher phase synchronization has been observed between the brains of parents and infants during direct eye contact[22]. During direct gaze, they also observed that the adult exerted a stronger influence on infant's neural activity, evidencing that eye contact might lead to stronger modulation or affect the direction of the synchronization.

Directed inter-brain synchronization has been observed in leader-follower scenarios[22,28–30], a phenomenon also demonstrated in non-human animals[31]. Another study[29] demonstrated that leaders presented stronger motor-related oscillatory patterns compared to followers when interacting in a finger-tapping task. A computational modelling study[32] explained this effect by demonstrating that successful behavioural interaction requires an increase in between-unit coupling (e.g., inter-brain) and a decrease in within-unit (e.g., intra-brain) coupling. For instance, they observed that leader-follower interactions require the follower to have low within-unit coupling whereas the relationship between two leaders tends to result in low between unit coupling. Taken together, these studies suggest that individual brains' responses might affect the dynamics of interactions, and vice-versa. These findings highlight the need to understand how interactions work in the dual brain system, combining both inter- and intra- brain connectivity. Since eye contact is a key factor in initiating and coordinating human interactions, it is important to determine if eye contact alone (a) plays a role in establishing leader-follower dynamics, and (b) results in directed synchronisation between brains, for instance, from leader to follower.

Graph theory can be used to quantify the properties of entire networks with measures that estimate how information flows through their nodes (i.e. brain areas) via their edges (i.e. connections)[33]. A few studies have exploited graph theory to understand the global and local characteristics of the so-called *hyperbrain* networks which include both intra- and inter-brain connections[34–36]. For example, a study[36] observed that the brain networks of an uncooperative dyad (two defectors in a prisoners' dilemma game) contained less interbrain links and were more modular (i.e. stronger connectivity within brains than between brains). Therefore, the current study aimed to investigate the hyperbrain networks during eye contact and to reveal their functional role by comparing the network properties of friends vs. strangers (i) and of spontaneously emerging leader-followers (ii).

Most hyperscanning studies report inter-brain synchronization during social interactions when people are face-to-face. Therefore, it is important to investigate the role of eye contact—a distinguishing feature of face-to-face interactions—in the hyperbrain dynamics, notably focusing on inter- and intra-brain synchronization. Here, we designed an experimental task which enabled us to isolate the role of eye contact to answer the following research questions: RQ1: How does eye contact affect inter- and intra-brain synchronization? RQ2: What are the network characteristics during eye contact? RQ3: What is the functional role of these networks? For instance, how do inter- and intra-brain synchronization during eye contact differ between friends and strangers? RQ4: Is the inter- and intra-brain synchronization during eye contact directed according to spontaneous leadership roles that emerge in the task? We hypothesised that eye contact would be associated with higher inter-brain compared to intra-brain synchronization and that the networks of friends would present higher number of interbrain connections. We also expected that the synchronization would be directed from leader to follower in dyads where leadership roles were clearly defined.

## Results

We designed an experimental task to isolate the eye contact from the other elements of the social interaction (Fig. 1b), by having participants make a duration reproduction task at the same time. While this task reduces the ecological validity, it was necessary because people do not naturally make uninterrupted eye contact without doing something else (e.g. talking). By giving them a time reproduction task, we tried to minimise the awkwardness of the eye-contact task whilst keeping some elements as close as possible to the characteristics of eye contact in real life. The time reproduction task also enabled us to measure inter-brain synchronization (EEG) during short bouts of eye contact similar in duration to those people usually engage in during a face-to-face interaction[37]. Considering individual differences in relation to preferred eye-contact duration, we chose time intervals of 1.5 s and 2.5 s which are within the durations found to be comfortable for people[38]. The use of two different durations also enabled us to test whether the participants were truly engaging with the task and whether they changed their estimations based on their partners (e.g. engaged in a leader/follower dynamics). The high time resolution of EEG enabled us to measure phase synchronization during these short bouts of eye contact.

Our behavioural results (Supplementary Note 1) showed that people engaged with the time reproduction task, reproducing lower durations following short intervals and longer durations following longer intervals (Supplementary Fig. 1a). Our behavioural results showed that: 1) participants underestimated durations during mutual eye contact compared to the control condition; and 2) during eye contact, the estimated durations changed according to their partner's estimations, which was evidenced by a correlation between the pair's estimations (Supplementary Fig. 1b). Furthermore, we found that in some pairs, one participant consistently gazed down first during the eye-contact condition, while the other participant followed, and this

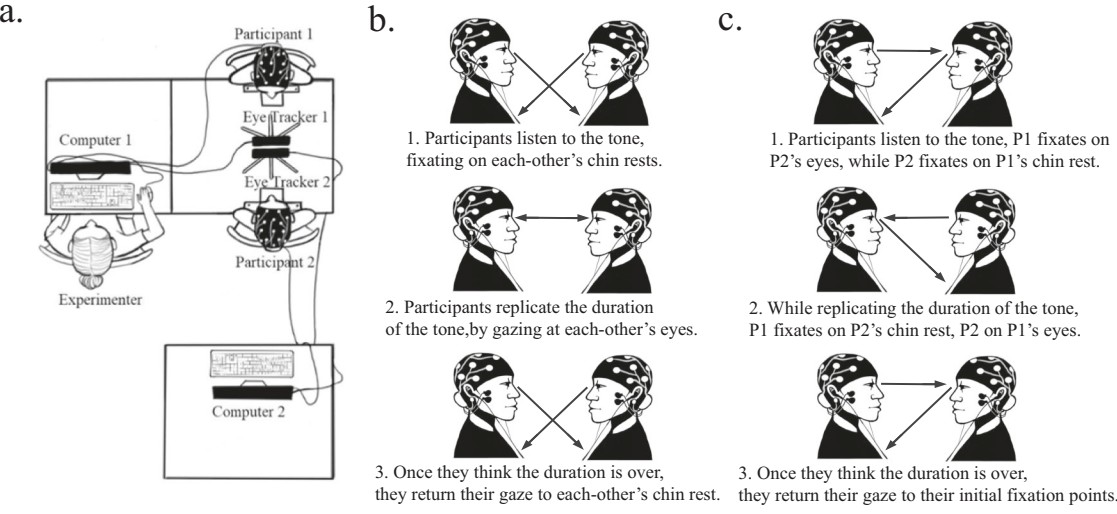

a.

Participant 1

Eye Tracker 1

Computer 1

Eye Tracker 2

Participant 2

Experimenter

Computer 2

b.

1. Participants listen to the tone, fixating on each-other's chin rests.

2. Participants replicate the duration of the tone, by gazing at each-other's eyes.

3. Once they think the duration is over, they return their gaze to each-other's chin rest.

c.

1. Participants listen to the tone, P1 fixates on P2's eyes, while P2 fixates on P1's chin rest.

2. While replicating the duration of the tone, P1 fixates on P2's chin rest, P2 on P1's eyes.

3. Once they think the duration is over, they return their gaze to their initial fixation points.

**Fig. 1 Experimental setup and task. a** Two computers were synchronized during the experiment and each computer collected EEG and eye-tracking data from one participant and received EEG and eye-tracking event markers with no delay (all centrally controlled through Matlab). **b** Eye-contact time estimation task: participants were required to reproduce the duration of a tone delivered through headphones without speaking to their dyadic partner. Eye-contact condition: during the delivery of the tone, participants were instructed to fixate on a sticker on the other participant's chinrest. After the tone ended, participants were instructed to look at the other's eyes for the duration of the tone, looking back down to indicate that they finished reproducing the tone duration. The tones were either long (2.5 s) or short (1.5 s). **c** No eye-contact control condition: participants replicated the duration of the tone but never made eye contact. Participant A listened to the tone while fixating on their partner's eyes while participant B listened to the tone while fixating on their partner's chinrest. After the tone, both participants replicated its duration, participant A by fixating their partner's chinrest (fixation point marked with a dot sticker) and participant B by fixating their partner's eyes. To indicate the end of the tone interval, each participant reverted their gaze back to the starting position. In one block, participant A replicated the duration by looking at their partners' chinrest and the other by looking at participant B's eyes (while participant B looked down to the chinrest), and in the other block, the roles reversed. The analysis of connectivity was restricted to the data from the period where both participants were performing the time reproduction task (step 2 of **a**, **c**). Drawing credits to Tatiana Adamczewska.

was not the result of one person simply responding earlier than the other (Supplementary Note 2). We considered the participant who gazed first to be the leader and the one who gazed second to be the follower, roles that were then used to investigate directed connectivity. The association between gaze following behaviour and leadership has been observed in both non-human[39] and human animals[40], and in this study, we investigated whether the direction of the synchrony between brains changes according to people's leader/follower roles.

We implemented a two-step approach for both undirected (corrected imaginary phase-locking value, ciPLV) and directed (phase slope index, PSI) synchronization measures. The ciPLV provides a robust measure of undirected phase synchronization which is insensitive to volume conduction, whereas the PSI provides a measure of non-instantaneous phase synchronization (small time delay), therefore directed (i.e. the phase of one signal precedes that of the other). We first compared the connectivity during eye contact vs. control task using a non-parametric cluster permutation approach (see Methods). This enabled us to identify a frequency band to conduct the network analysis. We then looked at the network characteristics considering both inter- and intra-brain connections as a single network, and examined if they differed between friends and strangers and if they were directed from leader to follower.

**Inter-brain undirected phase synchronization during eye-contact.** Our nonparametric cluster permutation analysis on the ciPLV revealed a significant cluster in the gamma frequency band (30–45 Hz) with higher inter-brain synchronization during eye contact compared to the control task (Fig. 2). The cluster has 54 significant links, mainly between the partners' right hemispheres (cluster *t*-statistic = 132.94, *t*-critical = 76.84, *p* = 0.0148). The topography of the inter-brain connections shows that the highest

differences in phase synchronization during eye contact compared to control were in the right hemisphere (including midline, Fig. 2b). The direct comparison of gamma synchronization in the identified cluster confirmed the presence of significantly higher phase synchronization during eye-contact compared to control ($t(49) = 4.282$, $p < 0.001$, Cohen's $d = 0.606$, Fig. 2c). There were no significant clusters in any other frequency band ($p > 0.05$).

To ensure that our effects were not caused by a difference in common sensory input between the conditions (as demonstrated by Burgess[41]), we created 1000 shuffled datasets for each condition (eye contact and control) where the data of participant 1 of the dyad was matched with participant 2 from another dyad in each condition. We hypothesised that if our gamma cluster was caused by a common sensory input, the positive cluster values (eye contact>control) obtained from such shuffled datasets would reflect this difference as phase synchronization would be higher in the shuffled eye contact compared to control. To test this hypothesis, we calculated the cluster statistics for each of these shuffled datasets (the procedures of the cluster permutation analysis were identical to the main analysis but instead of shuffling between labels, we shuffled the pairs as described above —pairing the data of participant 1 in one dyad with participant 2 from another dyad). We tested the significance of our real cluster statistics (our positive eye-contact cluster) against the cluster statistics distribution using the randomly shuffled participants (1000 different datasets). We found that the probability of finding the cluster we observed (real cluster t-statistic = 132.94) using shuffled data was very low ($p = 0.0020$, t-critical = 79.02).

**Intra-brain undirected phase synchronization during eye-contact.** We also looked at the differences in intra-brain phase synchronization during eye contact compared to the control task. A nonparametric cluster permutation analysis on the intra-brain

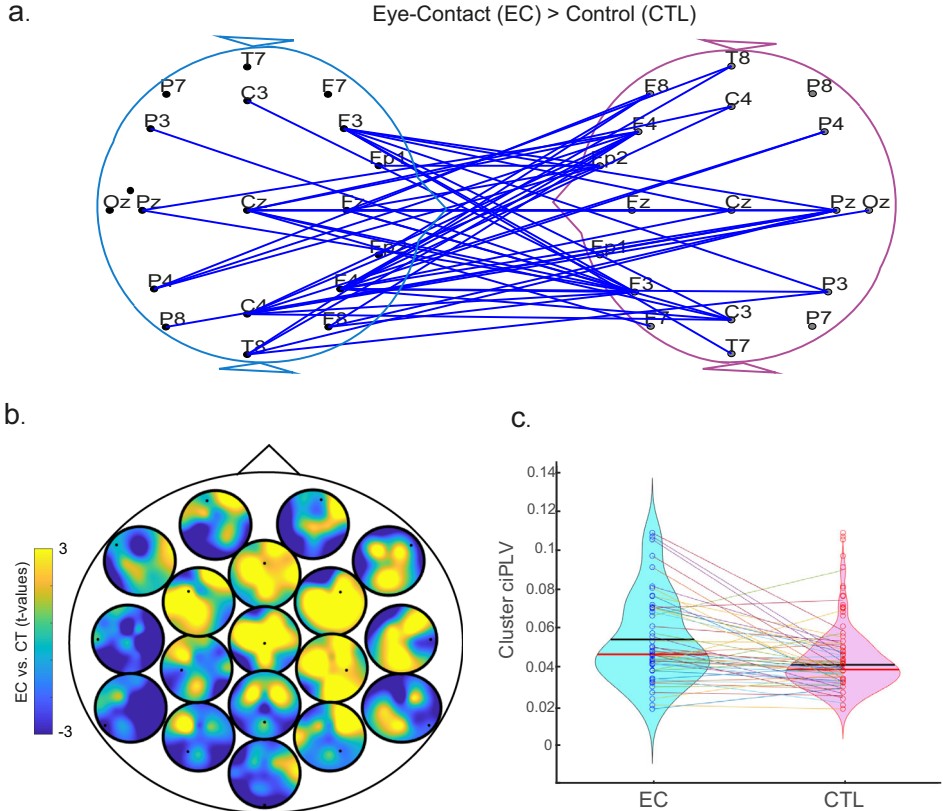

**Fig. 2 Interbrain synchronization during eye contact.** Gamma-band brain synchronization (ciPLV) within pairs during eye contact (EC) vs. control condition (CTL). **a** Blue lines represent electrode pairs with significantly higher gamma synchronization in EC compared to CTL. The significant cluster shows widespread connections between brains, especially distributed on the right hemisphere. **b** Heads-in-head representation of the difference in gamma synchronization during eye contact compared to the control task for the inter-brain connections. Each circle represents the scalp topography of the connections with the channel highlighted inside as a black dot. The colours represent differences in phase synchronization between that channel (dot) and the other person's scalp during eye contact expressed as a t-value of the contrast through the cluster permutation analysis. Yellow colours represent higher t-values for synchronization during eye-contact compared to control. **c** Violin plots showing the mean (red) and median (black) gamma synchronization for the cluster of interbrain connections (highlighted in blue in a) during eye contact and during the control task. Each datapoint is displayed in the figure and the lines show the changes for each pair—from eye contact (EC) to control (CTL).

ciPLV was conducted to compare the intra-brain synchronization networks between conditions. We observed a significant positive cluster (cluster $t$-statistic = 192.07, $t$-critical = 87.94, $p = 0.0123$) with higher synchronization in gamma frequency band during eye-contact (compared to control) with connections mostly on the right hemisphere (Fig. 3). The direct comparison of gamma synchronization in the identified cluster confirmed the presence of significantly higher synchrony during eye-contact compared to control ($t(99) = 3.676$, $p < 0.001$, Cohen's $d = 0.368$).

To ensure that the differences in phase synchronization (ciPLV) cannot be explained by differences in gamma power between conditions, we conducted a control analysis in which we compared gamma power (absolute and relative) between conditions in each of the measured channels (using the same data used for the ciPLV). Our results (Supplementary Note 3) showed no significant differences in gamma power during eye contact compared to control.

**Undirected network analysis: friends vs. strangers.** Since we observed that eye contact was associated with increased synchronization in the gamma band for both inter- and intra-brain connections, we analysed the global and local network properties of the gamma band networks using graph theory measures. To uncover the functional relevance of the undirected connections,

we compared the network characteristics of pairs of friends vs. strangers (see Methods for descriptions of all measures extracted from the network analysis).

For each pair, we calculated the z-scores of each edge based on the mean and standard deviation of gamma ciPLV in the control condition and thresholded the matrix for each pair. Despite having higher absolute values, inter-brain connections showed significantly higher z-scores than intra-brain connections ($t(99) = 5.623$, $p < 0.001$, Cohen's $d = 0.562$). This shows that, in general, making eye contact is associated with higher increases in synchronization between brains (interbrain) compared to the connections within a brain (intrabrain), as shown in Fig. 4a. To investigate the network characteristics relative to the baseline (control task), we calculated the following network measures: network strength (average z-score of edges), density, global and local efficiency, modularity and rich-club coefficient.

First, we analysed how friendship was associated with the differences in network strength and density between inter- and intra-brain connections. We applied a 2 (edge type: inter vs. intra brain) × 2 (friendship: friends vs. strangers) mixed-design ANOVA on the average network strength (mean z-scores of network edges). We observed that inter-brain connections increased significantly more during eye-contact compared to the intra-brain connections ($F(1,98) = 37.551$, $p < 0.001$, partial $\eta^2 = 0.277$). There was also a main effect of friendship

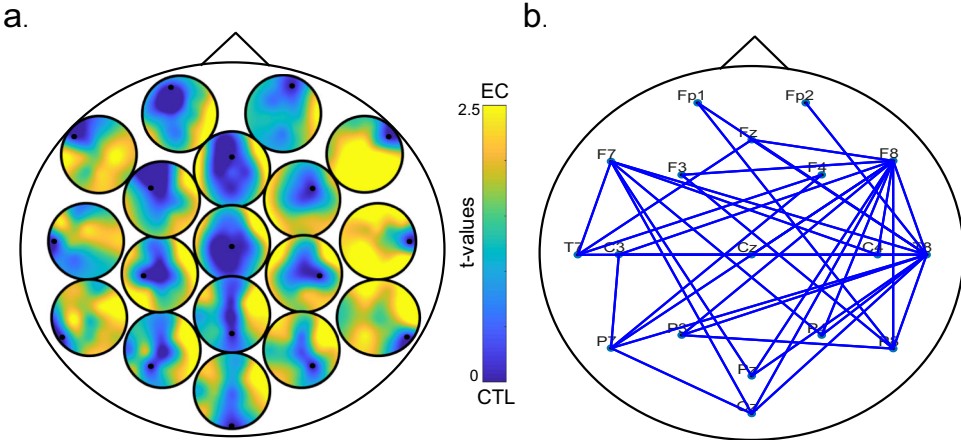

**Fig. 3 Intrabrain synchronization during eye contact.** Significant cluster of intra-brain connectivity during eye contact (EC) compared to no eye contact (CTL). **a** Heads-in-head representation of the difference in gamma phase synchronization during eye contact compared to the control task. Each circle represents the topography of the connections with the channel marked as a small dot. Yellow colours represent an increase in phase synchronization during eye contact expressed as a t-value of the contrast as conducted in the cluster permutation analysis. **b** Significant connections in the observed gamma frequency cluster. Most significant connections are between right parietal and the rest of the brain.

($F(1,98) = 9.567$, $p = 0.003$, partial $\eta^2 = 0.089$) and a significant interaction between edge type and friendship ($F(1,98) = 7.792$, $p = 0.006$, partial $\eta^2 = 0.074$). Pairwise contrasts showed that friends had significantly stronger inter-brain connections than strangers ($t(98) = 3.029$, $p = 0.003$, Cohen's $d = 0.608$), but not intra-brain connections ($t(98) = 0.199$, $p = 0.843$, Cohen's $d = 0.040$). Similarly, friends showed significantly stronger inter-brain compared to intra-brain connections ($t(45) = 5.593$, $p < 0.001$, Cohen's $d = 0.825$), which was also significant for strangers ($t(53) = 2.669$, $p = 0.010$, Cohen's $d = 0.363$).

We conducted the same analysis for density and observed a higher density for interbrain compared to intra-brain connections ($F(1,98) = 28.532$, $p < 0.001$, partial $\eta^2 = 0.225$), a significant effect of friendship ($F(1,98) = 6.627$, $p = 0.012$, partial $\eta^2 = 0.063$), and a significant interaction between edge type and friendship ($F(1,98) = 6.627$, $p = 0.012$, partial $\eta^2 = 0.063$) since the networks of friends showed a significantly higher inter-brain density compared to strangers ($t(98) = 2.955$, $p = 0.004$, Cohen's $d = 0.593$), but similar intra-brain density ($t(98) = 0.757$, $p = 0.451$, Cohen's $d = 0.152$). Inter-brain density was higher than intra-brain density for friends ($t(45) = 4.921$, $p < 0.001$, Cohen's $d = 0.726$) and strangers ($t(53) = 2.237$, $p = 0.030$, Cohen's $d = 0.304$). The averaged (and thresholded) connectivity matrix for friends and strangers (Fig. 4c) demonstrated a larger number of inter-brain connections between friends compared to strangers. These analyses reveal that making eye-contact affects the number and strength of the inter-brain connections more than the intra-brain ones, and that inter-brain synchronization during eye-contact is higher in friends compared to strangers.

Second, we compared the main network measures between friends and strangers (global and local efficiency, assortativity and rich-club structure). Networks of friends showed higher global ($t(48) = 2.664$, $p = 0.009$, Cohen's $d = 0.535$) and local efficiency ($t(48) = 2.192$, $p = 0.031$, Cohen's $d = 0.440$), but no difference in modularity and assortativity ($p > 0.05$, Fig. 4b).

To test if the networks have higher connectivity and efficiency in specific areas, we grouped the electrode data into 8 regions of interest (ROIs Fig. 4d): right frontal (RF: F4, F8, Fp2), right parietal (RP: P8, P4), left frontal (LF: Fp1, F7, F3), left parietal (LP: P3, P7), right centro-temporal (RCT: C4, T8), left centro-temporal (LCT: C3, T7) midfrontal (MF: Fz, Cz) and midposterior (MP: Pz, Oz). We averaged the degree (number of connections), and local and global efficiency for each ROI considering the entire network (intra and

inter-brain connections). We conducted a 2 (friendship: friends vs. strangers) × 8 (ROI: LF,RF,LP, RP, MF,MP,RCT,LCT) mixed-design ANOVA on each of these dependent variables (global and local efficiency and degree). Regarding global efficiency, we observed a significant effect of ROIs ($F(7686) = 34.789$, $p < 0.001$, partial $\eta^2 = 0.267$), a main effect of friendship ($F(1,98) = 6.044$, $p = 0.016$, partial $\eta^2 = 0.058$), but no interaction between ROIs and friendship ($F(7686) = 1.444$, $p = 0.215$, partial $\eta^2 = 0.015$). Pairwise comparisons (Fig. 4d) between ROIs showed that the midposterior region exhibited higher global efficiency than all the other regions ($p < 0.05$), followed by the midfrontal region which was also higher ($p < 0.05$) than all others (except midposterior). The frontal areas (left and right) showed the lowest local efficiency compared to all others ($p < 0.001$), but they did not differ between each other (all contrasts are Bonferroni corrected). We observed very similar results for degree and local efficiency since they were also higher in the midfrontal and midposterior regions and lowest at the frontal regions bilaterally. They were also higher for friends compared to strangers (main effect of friendship), and did not interact with ROIs, which suggests that friends and strangers showed a similar network structure (Supplementary Note 4).

We investigated whether the networks showed a rich-club configuration. We calculated the rich-club coefficient for each pair's network and considered the structure as rich-club (see Methods). We found evidence of rich-club structure in 43 out of the 50 pairs (86%). From all friends ($n = 23$), 18 pairs presented a rich-club structure (78.3% of the friend pairs) whereas 25 pairs of strangers (out of 27: 92.6%) presented such a structure (chi-square = 2.119, $p = 0.145$), revealing no significant difference in the rich-club structure between these groups. For each of the pairs who presented a rich-club structure, we calculated the proportion of times each node was a rich-club (Supplementary Fig. 4c) against all the pairs who showed a rich-club structure. This analysis showed that the key channels to present a rich club structure were the midline parietal (Pz), followed by the right and left parietal (P4/P3) and central (Cz, C4, C3). The channels which were less likely to be hubs were the frontal areas (bilaterally). These findings suggest that the midline and parietal regions might serve as hubs, which integrate internal and external connections during eye contact.

**Directed connectivity: leaders vs. followers**. We adopted the same approach to examine whether the connectivity was directed

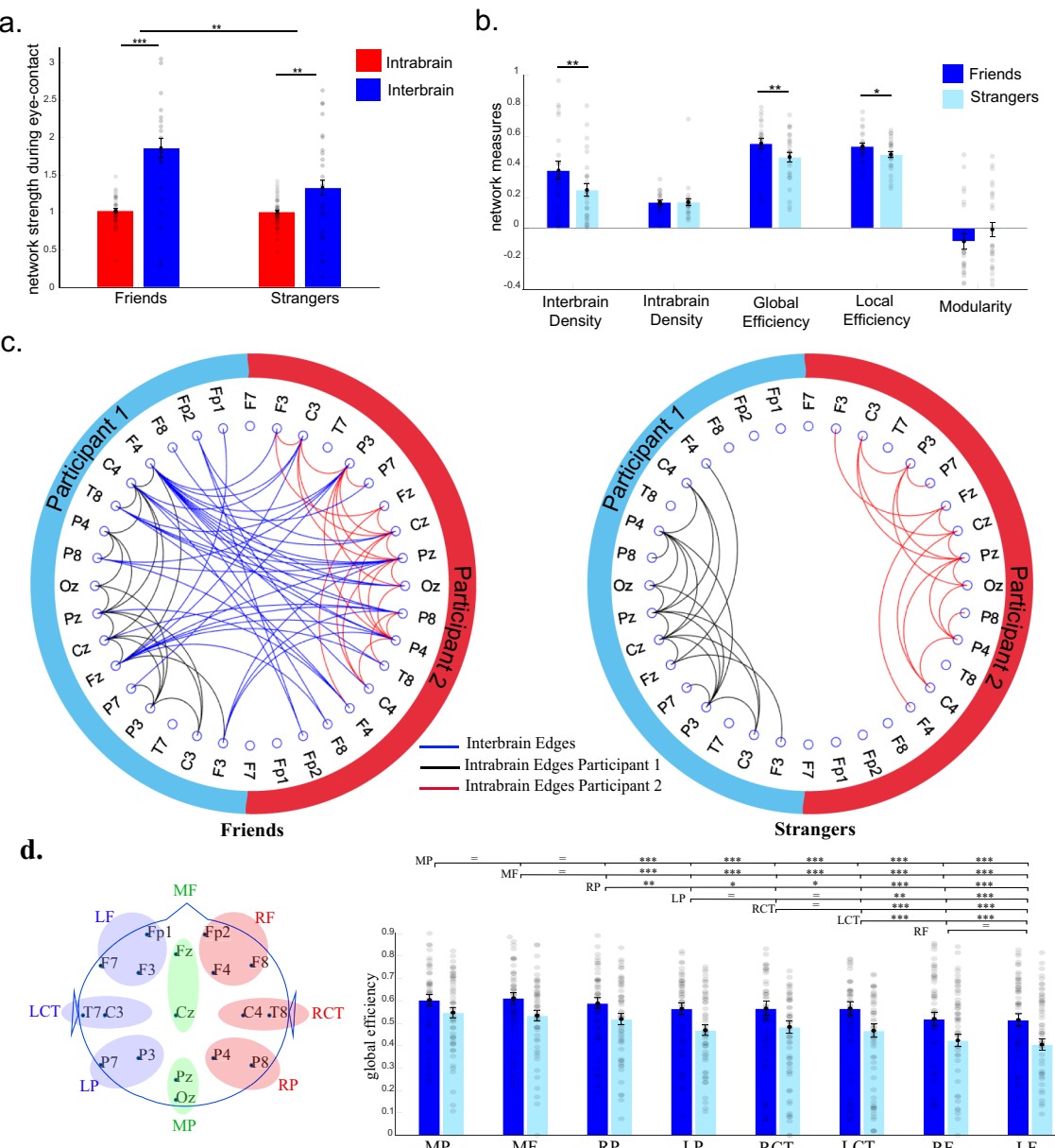

**Fig. 4 Hyperbrain network during eye contact. a** Intra- (red) and inter-brain (blue) network strength of friends and strangers measured as average of z-scores during eye-contact against control. The shaded dots represent each participant data ($n = 99$/ the data of one participant was not plotted because its value was 3.90, which would increase the y-axis range making the error bars illegible). **b** Network measures of friends (dark blue) and strangers (light blue) during eye contact. The measures include inter-and intra-brain density ($n = 100$), global and local efficiency and modularity ($n = 50$ dyads). These measures are based on the binary ciPLV networks. **c** ciPLV networks during eye contact, the edges represent the phase synchronization values which increased on average by more than 1 SD against the control task for friends (left) and strangers (right). Inter-brain edges are represented in blue, intra-brain edges of participants 1 are in black and 2 are in red. **d** Global efficiency of each ROI (left hand side): MP mid posterior, MF midfrontal, RP right parietal, LP left parietal, RCT right centro-temporal, LCT left centro-temporal, RF right frontal, LF left frontal. Pairwise contrasts are Bonferroni corrected. Error bars represent ±1 SEM. *** $p < 0.001$/** $p < 0.01$/ * $p < 0.05$.

during eye contact, between brains (e.g., from leader to follower) and within brains. As explained above, the phase slope index (PSI) indicates whether the phase of a signal precedes the phase of the other signal in each frequency band. If there is a leader-follower effect, we expect the phase of the leader to precede the phase of the brain signals of the follower.

*Inter-brain directed connectivity.* Our behavioural analysis revealed that in some pairs, one participant consistently gazed down first (the leader of the pair). For this analysis we only used pairs with clear evidence of leadership, i.e. strong leadership

(described in Supplementary Note 2). We adopted a nonparametric cluster permutation considering the direction of the connection (from leader to follower) separately and the condition which showed stronger connectivity values (eye-contact vs. control). This enabled us to use the same cluster analysis approach we adopted for analysing undirected phase synchronization. This approach enabled us to define the prominent frequency of the network, both for inter-brain and intra-brain connections.

We observed a significant inter-brain cluster in the alpha frequency band (Fig. 5a) during eye contact from leader to follower (cluster statistics $= 43.35$, $t$-critical $= 27.33$, $p = 0.0278$). Figure 5b illustrates

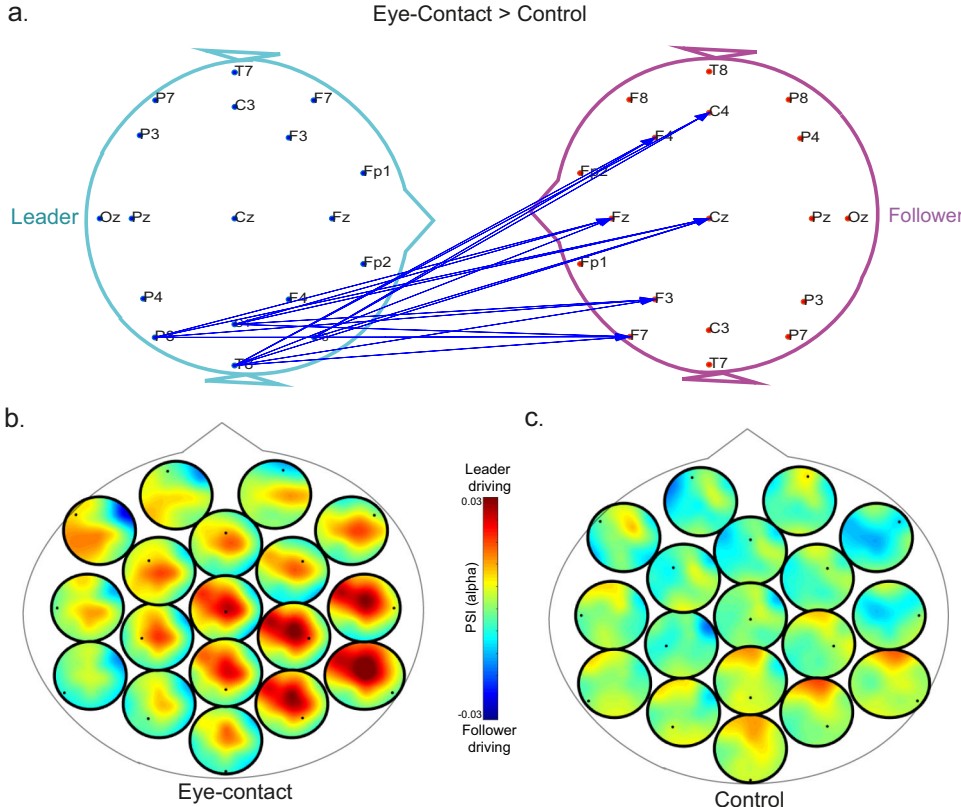

**Fig. 5 Directed phase synchronization cluster during eye contact between leaders and followers. a** Significant cluster connections from leader (left) to follower (right) in the alpha band. **b** Heads-in-head representation of the directed connectivity during eye-contact. Each dot represents the channel of the leader in relation to the follower. Red colours mean the channel is driving that area of the follower. For example, the right parietal electrode (P8) seems to be driving the activity of frontal and midfrontal of the follower (represented in red). **c** The same representation as in **b** but during the control task.

the full topography of all the directed connections from leader to follower during eye contact and during the control task. During eye contact (compared to the control task), the synchronization flux was from leader to follower, with the right hemisphere regions leading the frontal and midline areas of the follower. Directed phase synchronization in this cluster from leader to follower was positively correlated with leadership strength during eye-contact ($r = 0.508$, $p < 0.001$), but not during the control task ($r = -0.265$, $p = 0.066$). There was no cluster in the opposite direction (from follower to leader) nor clusters showing higher synchronization during the control task or in any other frequency band.

**Intra-brain directed connectivity**. We applied nonparametric cluster permutation to the directed intra-brain connections estimated using the phase slope index (PSI) in each frequency band. We followed the same nonparametric cluster permutation approach but taking into consideration the direction of the edges (see Methods). Consistent with the inter-brain findings, we observed a significant cluster in the alpha band showing higher synchronization during eye contact compared to control (cluster statistic = 15.737, t-critical = 10.157, $p = 0.033$), but not in the opposite direction (no cluster for the control > eye-contact task). The cluster shows that during eye contact, there was higher flux from frontal to parietal and occipital areas (Fig. 6a). Figure 6b shows a pattern where posterior regions, especially on the right are driven by frontal and left frontal regions. Clusters in all the other frequency bands were not statistically significant.

**Network analysis of the directed networks**. We extracted graph theoretical measures for each pair following the same procedures

adopted for the ciPLV based on the thresholded matrices. We extracted the same measures as in the previous analysis but now considering the bidirectional matrix based on the PSI (see Methods). To evaluate whether eye contact was associated with higher changes in intra vs. inter-brain edges, we compared the networks average strength and density. The results (Supplementary Note 5) demonstrated that eye contact was associated with higher increase in inter-brain connectivity compared to intrabrain during eye contact, independently of the leadership strength.

Our hypothesis was that the direction of the synchronization would be from leader to follower. To test this, we counted the proportion of outgoing connections compared to incoming from leaders to followers (Fig. 7a). We entered the proportion of outgoing edges (against incoming edges) in a 2 (leader vs. follower) × 2 (leadership strength: strong vs. weak) between-subjects ANOVA. We observed that the leaders presented a significantly higher proportion of outgoing connections compared to followers ($F(1,94) = 5.642$, $p = 0.020$, partial $\eta^2 = 0.057$) and a significant interaction with leadership strength ($F(1,94) = 13.500$, $p < 0.001$, partial $\eta^2 = 0.126$) since the leaders showed significantly higher number of outgoing connections compared to followers in strong leadership pairs ($t(40) = 2.946$, $p = 0.005$, Cohen's $d = 0.909$), which was not the case for the group with weak leadership ($t(54) = -1.622$, $p = 0.111$). To investigate the general effects of leadership on inter-brain connectivity strength, we compared the strength (z-scores) of outgoing (from leader to follower) vs. incoming (follower to leader) connections using a 2 (leaders vs. followers) × 2 (strong vs. weak leadership) × 2 (outgoing vs. incoming connections) mixed-design ANOVA. We observed (Fig. 7b) a significant three-way interaction ($F(1,94) = 9.010$, $p = 0.003$, partial $\eta^2 = 0.087$)

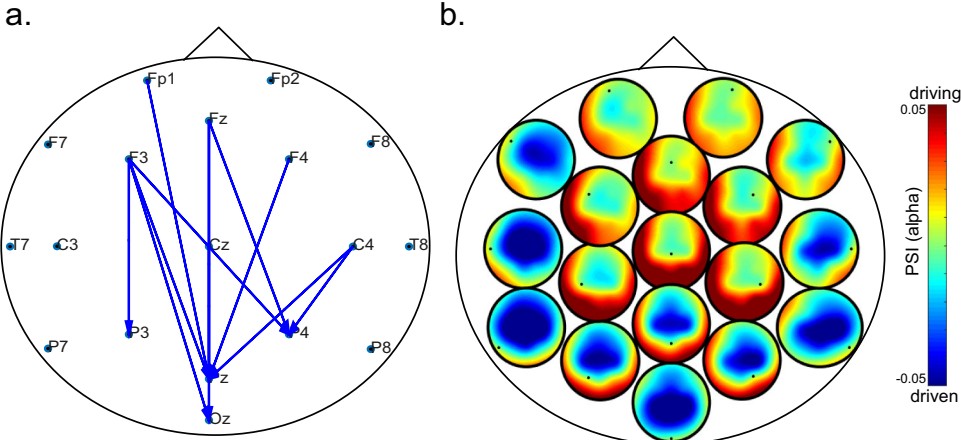

**Fig. 6 Directed phase synchronization intrabrain clusters during eye contact in the alpha band. a** The arrows show the significant connections for the cluster observed in the alpha band during eye contact compared to the control task (eye contact > control). **b** Heads-in-head plot showing the distribution of the PSI across the scalp for all electrodes. Each subtopography plot displays how the electrode represented as a dot (labels on **a**) relates to the rest of the scalp. For example, the central midline electrode (Cz) appears as the driver of the alpha phase of the posterior electrodes. Overall, there is a pattern of frontal regions driving posterior areas, which were higher during eye contact compared to control (**a**).

since there was a significant higher strength of outgoing connections from leaders compared to followers within pairs with strong leadership ($t(40) = 2.744$, $p = 0.009$, Cohen's $d = 0.847$), but not for the pairs with weak leadership ($t(54) = -0.944$, $p = 0.349$). As predicted, there was no difference between leaders and followers regarding the strength of incoming connections ($p > 0.3$).

We extracted the main network properties of each pair and compared them between pairs with strong vs. weak leadership. For these analyses, 1 pair was excluded because of lack of inter-brain connections. We did not observe any statistically significant difference between pairs with strong vs. weak leadership regarding their networks' global properties, including global efficacy ($t(46) = 1.238$, $p = 0.222$), average local efficiency ($t(46) = 1.067$, $p = 0.292$), modularity ($t(46) = 0.237$, $p = 0.813$), and assortativity ($t(46) = 1.200$, $p = 0.236$). Figure 7c illustrates the thresholded average networks of pairs with strong vs. weak leadership.

To investigate which regions in the network are more likely to serve as hubs, we calculated the number of connections (including all incoming and outgoing edges), global and local efficiency per channel, and averaged per ROI. We conducted a 2 (leadership strength: strong vs. weak) × 8 (ROI: LF, RF, LP, RP, RCT, LCT, MF, MP) × 2 (who leads: leader vs. follower) mixed-design ANOVA. We observed no significant effect ($p > 0.1$) except for a marginal trend towards a higher degree for participants in pairs with strong leadership ($F(1,92) = 3.121$, $p = 0.081$, partial $\eta^2 = 0.033$). The results showed that the number of edges was similar across ROIs and for leaders and followers, but slightly higher in participants in strong leadership pairs. To investigate the role of these ROIs in the networks, we extracted the global and local efficiency of each node and averaged across different ROIs. The global efficiency by node represents how much access to the whole network a certain node has. We conducted a 2 (leadership strength: strong vs. weak) × 8 (ROI: LF, RF, LP, RP, RCT, LCT, MF, MP) × 2 (role: leader vs. follower) mixed design ANOVA (Fig. S5). We observed no effect for ROIs, leadership and of role. However, there was a marginally significant interaction between leadership strength and role ($F(1,92) = 4.256$, $p = 0.042$, partial $\eta^2 = 0.044$) as the leaders of pairs with strong leadership showed higher global efficiency (Supplementary Note 6), especially on the midfrontal and right centrotemporal areas.

We conducted the same analysis using local efficiency as the dependent variable and observed no main effect or interaction between any of the factors ($p > 0.05$). Altogether these results suggest that the leaders' brains might have more access to the network (mainly through the medial and right posterior regions of the brain) than the follower's brains.

## Discussion

We investigated how eye-contact affects synchronization from within a brain to the hyperbrain, looking at both directed and undirected synchronization. By combining statistical and graph theoretical methods, we answered our 4 key research questions (RQs) and report the following key findings: 1) eye-contact was associated with higher connectivity between two brains compared to within a single brain, for both undirected and directed connections (RQ1); 2) eye-contact affects inter- and intra-brain synchronization in the same frequency: gamma band for undirected connectivity and alpha band for directed synchronization (RQ1); 3) the eye-contact hyperbrain network has a rich-club structure with hubs in the midline and parietal brain areas (RQ2); 4) friends making eye-contact have stronger inter-brain connections (RQ3); 5) inter-brain synchronization flows from leader to follower (RQ4); 6) the leaders' brain have more access to the entire network than the followers (RQ2, 3 and 4). These findings reveal key insights into the dynamics of eye contact as discussed below.

Eye contact involves a network of regions, most consistently those involved in social cognition such as the medial prefrontal cortex (mPFC)[42,43] with the anterior cingulate cortex[44], the superior temporal sulcus and fusiform gyrus[10,45], and the right temporoparietal junction (rTPJ)[46]. Looking into someone's eyes instead of the mouth was found to increase coupling between rapid and slow routes of visual processing, between the dorso-lateral prefrontal cortex (dlPFC) and regions involved in processing intentionality (posterior part of the superior temporal sulcus and medial prefrontal cortex) and within the social brain network[46]. On the other hand, hyperscanning studies using fNIRS observed that eye-contact increases coherence between two brains[12–14] and within brains[13]. A two-person neurofeedback study[24] using EEG in a museum showed that the amount of eye contact was positively correlated with coherence in alpha (9–11 Hz) and beta (26–30 Hz) – although gamma (>30 Hz) was

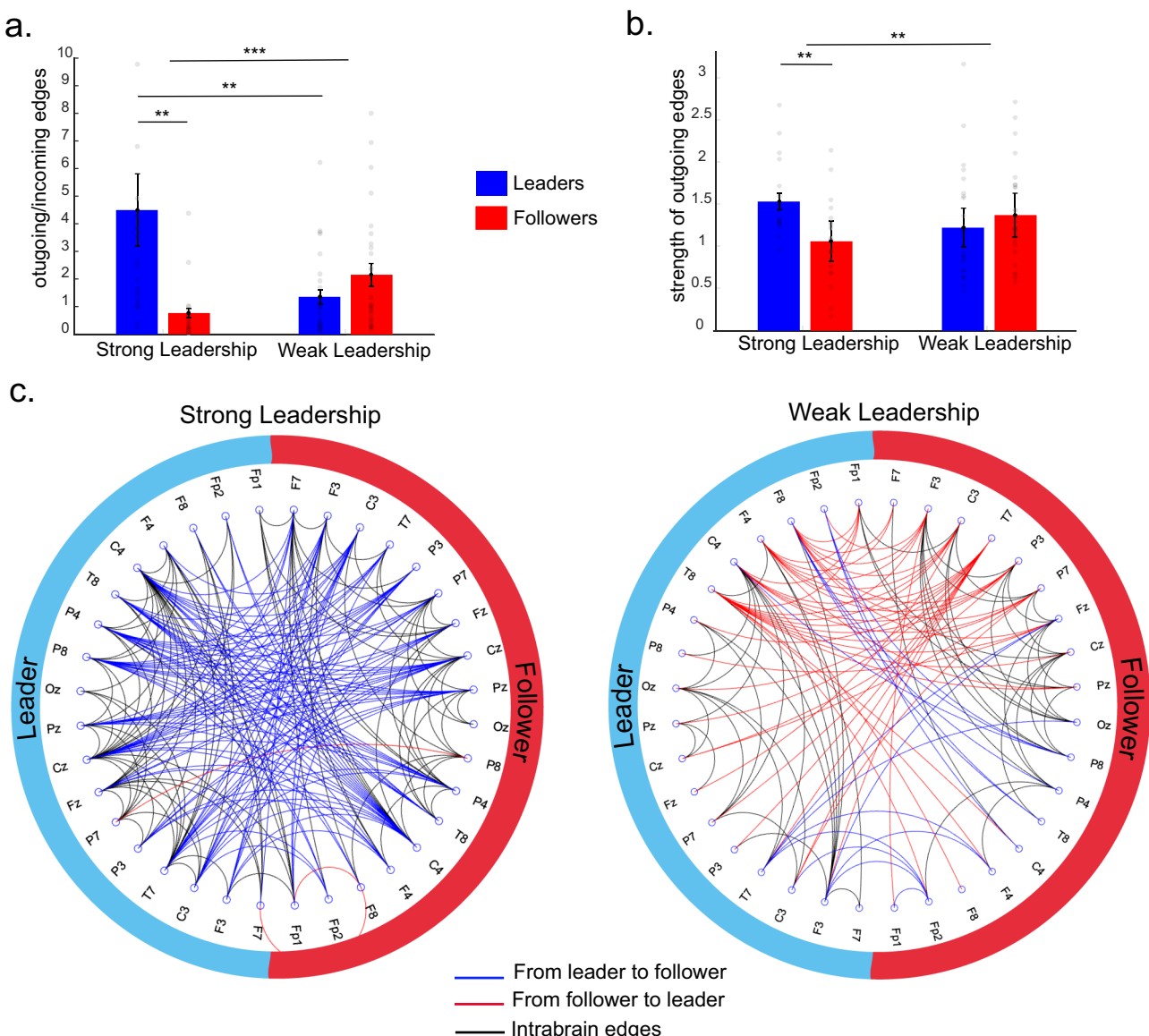

**Fig. 7 Directed connectivity (PSI) during eye-to-eye. a** Proportion of inter-brain outgoing in relation to incoming edges in leaders (blue) and followers (red) during eye contact in pairs with strong and weak leadership. The shaded dots represent each individual datapoint in each condition (*n* = 96 valid datapoints). Data for two leaders in a strong leadership dyad are not displayed to keep the data visible (datapoint 1 = 17.86/ datapoint 2 = 22.33). **b** Strength of inter-brain outgoing edges in leaders and followers during EC in pairs with strong and weak leadership (*n* = 96 valid datapoints). **c** Circular display of the intra and inter-brain networks during eye contact for pairs with strong (left-hand side) vs. weak (right-hand side) leadership strength. The blue edges represent connections from leader (left) to follower (right); the right edges represent connections from follower to leader; black represents intra-brain edges. These measures are based on the networks thresholded at 1 SD against the control task. Error bars represent ±1 SEM. *** *p* < 0.001/** *p* < 0.01/ * *p* < 0.05.

not tested. No studies on eye-contact have directly compared changes within and between brains' connectivity nor looked at the effects of eye contact on the wider hyperbrain network. We found clear and direct evidence that eye contact promotes higher changes in between-brains (inter-brain) synchrony compared to within (intra-brain), for both directed and undirected networks. These findings suggest that eye-contact is an inherently social and communicative signal. Critically, these increases in synchronization between brains during eye-contact are functionally meaningful: they were higher for friends compared to strangers and they were directed from leaders to followers. These findings support the interactive brain hypothesis[47–49] which posits that examining the brain alone is not sufficient to understand the synchronization we observe in social interactions. One of the

most contentious points of this hypothesis is that the synchronization between brains can be more informative of the nature of the social interaction than within individual brains. We observed that even though the intra-brain connections presented higher absolute values (due to the physical links and higher intra-brain correlation of EEG signals), the inter-brain connections were far more sensitive to eye-contact (i.e. social interaction).

We showed that eye-contact related phase synchronization (intra- and inter- brains) occurred in the same frequency depending on whether it was undirected (gamma) or directed (alpha) suggesting that intra- and inter-brain connections are part of the same network. Undirected connectivity was found in the highest frequency band while directed connectivity was observed in a slower frequency band, in line with the proposed functional

roles of these frequencies in the brain. There is evidence that top-down attentional processes involve lower frequencies while gamma synchrony reflects bottom-up attention[50]. There is also evidence that lower frequencies enslave or coordinate higher frequencies as demonstrated in cross-frequency coupling studies[51,52]. For instance, alpha was found to modulate neuronal firing in higher frequencies[53]; it was also found that high gamma power (80–150 Hz) is phase locked to theta oscillations[54]. Regarding the synchronization between brains, the suggestion is that top-down modulations would happen in lower frequencies since they would enslave higher frequencies. Here, our directed connectivity analysis revealed a lower frequency (alpha) than undirected synchronization (gamma) between the pair, supporting this hypothesis.

Previous EEG hyperscanning studies revealed increased synchronization during social interactions in a number of different frequencies, most notably in alpha (for a review on EEG hyperscanning studies, see[18,19]). For example, Dumas et al.[23] observed increases in phase synchronization between a model and an imitator in alpha, beta and gamma frequencies. Another study[55] found that effective social coordination was associated with an increase in synchronized alpha in the right centroparietal regions between a pair. Other studies[23,56–60] observed increased inter-brain synchrony in higher frequencies. A difficulty in reconciling these frequency differences is that the activities that the pairs perform vary largely between studies, making it harder to disentangle activity reflecting social coordination and task constraints[15], especially considering the issues with shared common input[41]. In our experiment, we evaluated synchronization during fixations (when both participants eyes were still), which reduces the influence of joint actions/movement on brain synchronization. Considering that eye contact leads to synchronized eye movements or mimicking behaviour[44], and that gamma-band activity was found to be involved in several social skills such as bonding[56], empathy[61], mentalizing[62], cooperation[60], and prosocial behaviour[58], it is possible that the gamma activity involved in processing social cues resets and aligns once eye-contact is established and participants engage in eye-mimicry. For instance, microsaccades happen within the gamma range (their approximate duration is 30ms[63]) and were found to induce bursts of high-frequency (gamma) neuronal activity along the visual pathways[64,65]. Therefore, it is possible that the eye-mimicry is what enables their fast gamma activity to align in phase, which could facilitate the processing of social cues. This is a hypothesis which deserves further investigation, especially in studies with high sampling frequency eye-tracking. This process of resetting gamma phase based on microsaccades mimicry could account for the greater synchrony between friends, since it is possible that the frequent exposure helps friends learn each other's patterns of movements or microsaccades which could facilitate mimicry and ultimately lead to a synchronous phase reset of gamma band activity.

We also found that leader-to-follower-directed synchrony occurred in alpha, which is consistent with the physiological interpretation of lower frequencies enslaving higher ones. Although we did not test cross-frequency coupling between the partners, it is possible that the leader's rhythms enslaved the higher frequency eye-contact processes of the follower. Here we suggest that simultaneous synchronization between participants reflects an increase in synchronization in higher frequencies while directional interactions will be reflected in synchronization in lower frequencies.

An alternative explanation is that different types of information are communicated in different frequencies. During eye contact, we are extracting a wealth of information about the person we are gazing at, including eye movements, emotions,

aesthetics, etc. and it is possible that the eye movements mediating mimicry are encoded in a different frequency compared to, for example, the reading of the others' emotions/intentions/expressions. To answer this question, new studies designed to isolate each of these components are needed.

Our leadership findings warrant further consideration as they are limited to our experimental task. The leadership behaviour emerged naturally, without any instruction to do so, as it is often the case with the emergence of leader/follower roles in naturalistic settings (e.g.[66,67]). Our study provides preliminary evidence that inter-brain synchronization direction during eye-contact might predict the leadership roles: with synchronization flowing from leader to follower. We also show that in the hyperbrain, the leaders seem to have more access to the entire network, which deserves further consideration. It would be interesting to understand whether the direction of this association is limited to the task at hand or whether it generalizes to other tasks. Our study was limited to this specific time-reproduction task, so we cannot speculate on whether directed alpha synchronization during eye contact would predict leadership roles in other tasks, such as dance or even in conversation.

We observed that the midposterior, midfrontal and right parietal regions may serve as hubs of the eye-contact network, in both directed and undirected interactions. One of the few studies looking at inter-brain synchronization during eye-contact[13] observed an increase in connectivity within and between brains during eye contact compared to looking at the eyes in a photo of a face. The authors observed an increased in connectivity between left frontal regions of one participant to left temporoparietal areas of the other. However, it is unclear if the increase in inter-brain synchronization and its lateralization were associated with the difference between gazing at a live face vs. a still photo as it has been demonstrated that live mutual eye contact involves different neural mechanisms than those involved in delayed off-line eye-contact[44]. For instance, eye contact has been shown to be associated with increased activity in the rTPJ compared to looking at a video of a dynamic face[14]. An increase in inter-brain rTPJ coupling was also found by a study[12] looking at eye contact during live interactions. While we cannot know the precise neural sources of this effect due to our limited spatial resolution, we can generate predictions regarding the dynamics of the interactions based roughly on the locations of these areas on the scalp and the previous literature on eye contact and social interactions. An insightful review[68] pointed out that hyperscanning studies often find that multiple brain regions synchronize during social interactions. The authors suggested that it is important to investigate the extent to which inter-brain dynamics vary across different regions. We provide evidence that some regions might play a key role in the eye-contact network by integrating external and internal online information about the self and the partner. The midline regions we found might reflect the activity of the medial prefrontal cortex and the posterior cingulate cortex, both along the midline. The mPFC and the PCC were both found to be involved in mentalizing[69,70]. The mPFC was found to control the level of mimicry driven by eye contact by modulating sensory processes in the superior temporal sulcus (STS)[43]. There is evidence that while regions of the mPFC and ventromedial PFC are biased for self-referential processing, the PCC and/or the precuneus, and the right temporoparietal junction are biased for other-referential processing[71–74]. Here we suggest that these areas, the medial frontal, medial posterior, and the right parietal may serve as hubs, which integrate information about the self (internal connections) and the other (represented as the inter-brain connections) during eye contact, enabling inter-brain synchronization.

Finally, we observed that effective connectivity flux went from leaders to followers. Previous studies have observed similar leader-follower-directed connectivity during music playing[30,75]

and group discussions[28]. It has also been observed[29] that during a leader-follower finger tapping task, the leader showed stronger frontal alpha desynchronization than the follower. Building on this idea, a computational modelling study demonstrated that in leader-follower interactions, the leader shows the highest within-unit coupling (in our case intra-brain) and the follower the lowest within unit coupling[32]. We did not find evidence of differential intra-brain coupling between leader and follower in our study, which might be because our paradigm did not require mutual and continuous adaptation between leader and follower. Instead, we found that the leaders' brain regions had more access to the entire network of leader-follower (i.e. hyperbrain). This means that the nodes of the leader's brains have shortest paths through the entire network, increasing its accessibility. Future studies could investigate how this can affect the amount of control that the leader's brain exerts in the dyad's behaviour in mutually interactive tasks.

Collectively, our findings support the hypothesis that eye-contact affects the synchronization between two brains more than it affects the links within each brain. They showed that the brains of friends synchronized more strongly but with similar network characteristics. Nonetheless, it is important to interpret some of these findings with caution. First, we looked at a small range of network measures (all reported) without correcting for multiple comparisons. It is possible that type I errors could occur even though most of our p values are not borderline. We did that in order to avoid type II errors and other known issues with Bonferroni corrections[76]. We suggest that our results to be interpreted as a pattern, which shows that eye contact is associated with higher inter-brain synchronization in friends (a result which was confirmed by all measures, such as network strength, degree, density, and efficiency), but not with specific differences in network characteristics such as modularity and assortativity.

Despite our attempts to include the key characteristics of eye contact in social interactions in our task design, replicating its short durations and including a secondary task, our study has limited ecological validity. In naturalistic settings, eye contact often emerges spontaneously[77] and it is held for the duration that both people are comfortable with[38], whereas in our study, eye contact was established on demand as a part of the experimental task. A study[77] observed that during conversation eye contact is initiated as pupillary synchrony peaks and it predicts immediate subsequent decline as it breaks. Although our findings do not enable us to speculate about the precursors of establishing or terminating eye contact, they could guide future analysis looking at the time course of interbrain synchronization during eye contact in naturalistic settings, especially regarding how undirected gamma and directed alpha networks change as eye-contact initiates, evolves, and terminates. It is of interest to find how these networks relate to changes in pupillary synchrony in naturalistic settings and how each of these components might be associated to autism spectrum disorders.

## Methods

**Participants**. Fifty-six adult pairs of participants (112 participants) took part in this experiment, all neurologically healthy adults. Of these, 27 pairs were friends (37 female, 17 male) with a mean age of 20.52 (SD = 1.59) years, while 29 were strangers (45 female, 13 male) with a mean age of 20.30 (SD = 2.30) years.

**Ethics**. All participants received a monetary compensation of £7.50 per hour for their participation and gave written informed consent before the beginning of the experiment. The study protocol was approved by the Queen Mary University of London ethics committee (QMERC1947). Experiments were conducted in accordance with the World Declaration of Helsinki (1964).

**Procedure**. Participants arrived in pairs and sat opposite to one another in a quiet lab room. First, the experimenters prepared the EEG equipment on both participants, so that each was connected to an EEG machine (Fig. 1a). Subsequently, an eye tracker was placed in front of each participant on a table (Fig. 1a) and angled to capture their respective eye movements. Each participant used a chinrest that was at the same height for both partners. The eye trackers were calibrated with the following procedure. To calibrate the eye-tracking of the participant on seat 1 (P1), the participant on the opposite seat (P2) held a calibration board with numbers 1–9 written on a 3 * 3 grid. P1 was instructed to look at each number for 5 s, while the positions of their eyes were registered on the computer and then the same was repeated for P2. Participants' eye movements were recorded at 60 Hz resolution. Earphones were provided and the sound volume was adjusted at a comfortable level. In all tasks, participants were presented with 1000-Hz tones, which were either 'short' (1.5 s) or 'long' (2.5 secs) in duration. There were 4 conditions that were presented randomly across trials 1) P1 and P2 both heard a short tone; 2) P1 and P2 both heard a long tone; 3) P1 heard a short and P2 a long tone; and 4) P1 heard a long and P2 a short tone. The order of the tasks (eye contact and control) was counterbalanced across pairs. We adopted these two different durations in order to: 1) ensure that the participants engaged with the time reproduction task as we can check this by comparing how they performed under each duration; 2) analyse a possible interference in behaviour manifested as a change in estimation depending on their partner's duration. For instance, if one partner hears a short tone, would their estimations be longer if their partner heard a longer duration?

**Eye-contact task**. Participants were instructed to fixate on a black sticker positioned on their partner's chin rest. They were then required to pay attention to the duration of the beep that would be presented to them through the earphones. When the beep ended, they were required to look at their partner's eyes for the duration of the beep they had just heard (they were asked to try to estimate the duration of the beep and reproduce the same duration). To indicate the end of their estimation, they looked back at the sticker until the next trial (Fig. 1b). The first 45 pairs were presented with 56 trials in total (14 trials in each of the four conditions: short-short, long-long, short-long, long-short tones), while the number of trials was doubled for the last 11 pairs to increase statistical power (112 trials in total).

**Control task**. To compare brain synchrony during eye contact vs. no eye contact, we conducted a control task where participants were required to look at their partner's eyes when the latter was looking at the chinrest, and vice versa (Fig. 1c). For instance, P1 was required to fixate on P2's eyes until the tone was presented. After the tone finished, P1 looked at P2's chinrest-sticker for the duration of the beep and returned to looking at the eyes once they had finished reproducing the duration. The other participant was instructed to do the opposite, first gaze down while listening to the beep and then reproduce the tone duration by staring at their partner's eyes. Therefore, in these control trials, the pair never made eye contact. The condition was repeated reversing the roles. Pairs were presented with trials in each of these two control blocks, 56 trials in total. The number of trials was doubled for the last 11 pairs. Trials were randomized across conditions. The order of the eye contact and control tasks was counterbalanced across pairs.

**EEG recording and preprocessing**. The EEG signals were recorded using two Starstim 20 (Neuroelectrices) EEG devices. We used eighteen PiStim electrodes placed according to the extended 10–20 electrode placement system (Jasper, 1958). The EEG electrodes were: P8, F8, F4, C4, T8, P4, Fp2, Fp1, Fz, Cz, Pz, Oz, P3, F3, F7, C3, T7, and P7. EEG data were re-referenced to the algebraic mean of the right and left earlobe electrodes[78]. Continuous data were high-pass filtered at .5 Hz and low-pass filtered at 45 Hz. Data from electrodes with poor signal quality, as observed by visual inspection, was interpolated from neighbouring electrodes. The data was epoched according to the onset of the common tone-reproduction period. Specifically, the start of the epoch corresponded to the time when both participants started reproducing the tone duration. The offset of the epoch corresponded to the time when one of the participants finished reproducing the tone. Independent component analysis could distort the phase of the signal which would in turn affect connectivity. We detected eye blinks automatically by identifying the time points of the signal when Fp1 and Fp2 electrodes had amplitude exceeding +/−70 μV, and excluded +/−0.040 sec from the data of all electrodes from subsequent analysis. The preprocessing was done using EEGLAB toolbox[79]. One pair was excluded from the EEG analysis due to technical issues in the EEG recording, while five more pairs were excluded due to insufficient number of valid trials (<5 trials per condition) ascribed to motor artefacts (N = 100, 50 pairs).

**Hyperscanning setup and synchronization**. The hyperscanning set up consisted of two connected and synchronized desktops via a crossover ethernet cable. Each desktop was connected to a different EEG and eye-tracking device. All processes were centralised in Matlab, which was used to stream the eye-tracking data in real time and send triggers to the EEG (via MATNIC toolkit using LSL) based on both the eye-contact task and the eye-tracking behaviour (e.g. saccades detection). During the experimental tasks, both computers streamed data from the eye-tracking machines in real time via Matlab, sending triggers for every event of interest, including saccade detection. The triggers were sent to both EEG machines simultaneously via TCP/IP communication. To test timing precision, we compared the timings (time stamps and samples) when each trigger was received by the EEG machines and the number of samples between two events (e.g. eye-movement/

saccade markers) in each of the two files recorded simultaneously. We found no discrepancy, showing that the timings were accurate and that the machines were fully synchronized.

## Data analysis

*Tone-reproduction duration.* We compared participants' gaze durations during tone duration reproduction. Specifically, we calculated the mean tone-reproduction duration of each participant (i.e. offset minus onset of eye movement during reproduction of the tone duration), separately in each task and in each condition. As we wanted to investigate whether the tone-reproduction duration of one participant would be influenced by their partner, a 2 (task: eye-contact vs.control) × 2 (tone duration: short vs. long) × 2 (pair duration: same vs. different) repeated measures ANOVA was conducted. We note here that the same pair duration corresponds to the trials when both partners heard a short or a long tone, whereas the different pair duration corresponds to trials when one partner heard a short tone and the other partner heard a long tone. Gaze durations of <0.5 or >4 secs were excluded.

*Behavioural analysis of the interaction between partners.* To test whether the partners were truly interacting during the eye-contact condition, we tested the correlation between their tone-reproduction durations. If the behaviour of one participant affected their partner, we would expect that they would make similar duration estimates when presented with tones of the same duration. We only used trials when both partners were presented with short or long tones to avoid inflating the correlations due to different tone durations. Pairs with less than 5 trials were excluded from the analysis. We expected that if the participants were truly interacting: 1) the time estimations of short and long intervals would change if their partner heard to a different duration; 2) the partners' time estimations would be correlated (Spearman correlation within trials) compared to the shuffled distribution of the same trials; 3) these effects would be stronger in the eye-contact condition.

First, for each pair, we tested the correlation between their tone-reproduction durations on trials where both participants heard the same duration in each condition (eye-contact vs. control). Second, since there was still a small degree of interaction in the control task (participants might have been able to see their partner's behaviour), we compared the correlation between the intervals against a shuffled distribution. For each participant, we shuffled the order of the trials (we did not shuffle between participants) and tested the correlations between partners for 5000 iterations. We then extracted the mean and standard deviations of the random distribution for each pair. Finally, we investigated whether their actual correlation value was higher from chance by statistically comparing the correlations obtained from the real data with the average correlations from the randomly shuffled data. These analyses are reported in the Supplementary Note 1 and 2.

*Undirected brain synchronization during eye-contact. Corrected Imaginary part of the PLV (ciPLV):* To compare brain synchronization between the EEG signals of the partners during eye-contact vs. control (no eye-contact), we calculated the ciPLV, a non-directed measure of phase synchronization[80]. The ciPLV is an optimized implementation of the Phase Locking Value (PLV)[81]. The PLV measures the instantaneous phase difference between two signals and considers that the signals are synchronized if they evolve together, i.e. if the phase difference is constant they are said to be locked. One of the issues with its original formulation is that the PLV is sensitive to volume conduction or zero-lag correlation. The ciPLV, besides being faster, uses the imaginary part of the PLV which removes the contributions of the zero phase differences. We followed the steps described by Bruna, Maestro & Pereda[80]: we first band-pass filtered the data within each of the frequency bands to extract the instantaneous phase of the signals: theta (4–8 Hz), alpha (8–12 Hz), lower beta (13–20 Hz), upper beta (21–30 Hz), and gamma (30–45 Hz). After this, we obtained a band-pass version of the Hilbert analytical signal which was then used to estimate the ciPLV using the code provided in the paper[80]. We analyzed the time window 0.5–2 sec after the onset of each epoch to avoid contamination with eye-movements. Trials with shorter durations were excluded from the EEG analysis.

*Brain synchronization during eye-contact vs. control task. Nonparametric cluster permutation test:* As there was no solid assumption to justify a hypothesis-driven analysis and considering the multiple comparisons problem, we used nonparametric cluster permutation approach[82] to compare the synchronization of oscillatory activity during eye contact vs. control task. To eliminate potential biases introduced by multiple comparisons and distribution assumptions of parametric tests, the difference distribution for eye-contact vs control networks was constructed in a data-driven manner using label randomizations combined with a network-based clustering criterion for the t-statistic extraction[83]. The network-based statistic controls for family-wise error rate offering a substantial gain in power by considering the topological characteristics of the graph assuming that a biologically relevant effect on the network cannot be isolated to single or disconnected edges. Meaningful clusters need to show strongly connected components (connected to each other). We first calculated the statistical difference for each brain edge (eye-contact vs. control), discarding absolute t-values lower than 2.

Then, the survived edges were clustered in strong connected components (SCCs; partition into subgraphs with the property of having at least one path between all pairs of nodes) depending on whether they reflect identical effects (separate clusters for positive and negative edges). Subsequently, difference distribution curves of the condition differences were estimated using 5000 permutations per frequency band by randomly shuffling the condition labels, respecting each pair's (for inter-brain tests) or each participant's (for intra-brain tests) data. In each iteration, we computed the sum of t-scores within each cluster and kept the maximum (absolute value) cluster score as the cluster t-statistic. The t-critical values were then calculated to align with the significance level of 0.05 (two-tailed). Clusters formed by the actual labels with t-score exceeding the t-critical values were finally identified following an SCC-wise inference on the difference distribution. This approach was used in all cluster analyses presented in this paper, both intra and inter-brain.

*Effect of leadership on directed brain synchronization.* To investigate how inter-brain synchronization is influenced by leadership roles, we first identified the leader and the follower in each pair. Then, we compared directed brain synchronization in pairs with strong vs. weak leadership patterns. Therefore, in each trial, the person who broke eye-contact from their partner first was considered the leader. We divided the number of trials in which participant 1 (P1) broke eye contact first compared to the total number of trials. The resulted values spanned from 0 (P2 leads) to 1 (P1 leads). Values around 0.5 meant the absence of a leader (P1 and P2 broke eye-contact in an approximately equal number of trials). As values around both extremes are indicative of strong leadership (e.g., 0.2 signifies the same leadership strength as 0.8; in the first case P2 is the leader, whereas in the second case P1 is the leader), we subtracted all values that were lower than 0.5 from 1. This resulted in values ranging from 0.5 to 1, with higher values indicating stronger leadership pattern. We used trials when both participants heard tones with the same duration (both short or both long).

We then split the pairs into two groups based on their leadership strength (median split) and found that only one pair displayed no leadership relationship between partners. This pair was excluded from this analysis, leaving 49 pairs in total. Pairs with strong leadership relationship (higher than the median leadership strength at 0.667) were considered as the high leadership dyads ($N = 26$), whereas pairs with weak leadership relationship were considered the low leadership dyads ($N = 23$).

*Phase slope index (PSI):* Directed inter-brain synchronization was measured using the phase slope index (PSI)[84] to estimate the synchronization between the electrodes of leaders and followers. If PSI from electrode X (leader) to electrode Y (follower) is positive this would mean that the leader is leading the brain synchronization. The PSI estimates the synchronization between two signals based on the slope of the phase of their cross-spectrum, and is insensitive to volume conduction while detecting non-instantaneous functional relations between two signals. In Nolte et al. (2008), PSI is defined as:

$$\widetilde{\Psi}_{ij} = \Im\left( \sum_{f \in F} C_{ij}^*(f) C_{ij}(f + \delta f) \right) \quad (1)$$

where:

$$C_{ij}(f) = \frac{S_{ij}(f)}{\sqrt{S_{ii}(f) S_{jj}(f)}} \quad (2)$$

is the complex coherency between sources $i$ and $j$, $S$ is the cross-spectral matrix, $\delta f$ is the frequency resolution of the coherency, and $\Im(\cdot)$ denotes getting the imaginary part. $F$ is the set of frequencies over which the slope is summed. The equation is rewritten as follows to see that the definition of $\widetilde{\Psi}_{ij}$ corresponds to a meaningful estimate:

$$\widetilde{\Psi}_{ij} = \sum_{f \in F} a_{ij}(f) a_{ij}(f + \delta f) \sin\left(\Phi(f + \delta f) - \Phi(f)\right) \quad (3)$$

with $a_{ij}(f) = \left| C_{ij}(f) \right|$ being frequency-dependent weights. For smooth phase spectra, $\sin(\Phi(f + \delta f) - \Phi(f)) \approx \Phi(f + \delta f) - \Phi(f)$ and hence $\Psi$ corresponds to a weighted average of the slope.

Finally, $\widetilde{\Psi}$ is normalized by an estimate of its standard deviation:

$$\Psi = \frac{\widetilde{\Psi}}{std\left(\widetilde{\Psi}\right)} \quad (4)$$

with $std\left(\widetilde{\Psi}\right)$ being estimated by the Jackknife method.

*Nonparametric cluster permutation on PSI networks:* we employed the same approach described above. However, since the PSI is directed, we calculated the clusters (SCCs) separately considering whether the connections were positive or negative and whether they were higher during eye contact or control. This avoided mixing up connections that belonged to different effects.

*Network Analysis.* We transformed the connectivity matrices (both ciPLV and PSI) for each pair based on their z-scores against the control condition. We calculated

the mean and standard deviation during the control task for inter- and intra-brain connections to use as a reference to calculate the z-scores for each connection during eye contact. Those matrices were thresholded at >1 SD (of the control) in order to extract the network measures. For the PSI matrices, the thresholded matrices also contained a signal (0, −1,+1). These matrices were used to calculate the graph-theoretical measures, including network efficiency (local and global), modularity, assortativity, and the rich-club coefficient. Additionally, we modified the efficiency measured to allow a measure of global connectivity of a node rather than the whole network. This measure enabled us to estimate how much access any given node has to the entire network.

*Phase synchronization thresholding for hyperbrain analyses:* For each pair $k$ of participants with a sufficient number (>5) of trials in both the eye-contact and control tasks, we computed the Z-scores for the synchronization measured between the electrodes' signals. For each possible intra-brain or inter-brain edge $(i, j)$, the synchronization Z-score at a given frequency band $f$ was computed as:

$$Z_{i,j}^k(f) = \frac{C_{i,j}^k(f) - \mu_{i,j}^k(f)}{\sigma_{i,j}^k(f)}, \tag{5}$$

where the mean value $\mu_{i,j}^k(f)$ and the standard deviation $\sigma_{i,j}^k(f)$ were computed individually across all the edges $(i, j)$, during the control for the $k$ couple (for interbrain edges, $i$ indexes P1 nodes and $j$ indexes P2 nodes). Since we noticed higher coherence values in intra-brain vs. inter-brain connections, we adopted two separate thresholds for the two types of edges, whereas for the intrabrain edges, Z-scores were calculated in the single-subject level to balance between the count of intrabrain connections within a particular pair.

*Hyperbrain networks from Z-scores:* Single-pair Z-transformed connectivity matrices were further processed to generate both an unweighted and a weighed graph representation. Using both the Z-thresholds of P1 and P2, we obtained (i) an unweighted graph from each single-pair connectivity matrix by binarizing the z-scores above and below the Z-threshold of 1 and (ii) a weighted graph version of the same data simply by ditching the below-threshold connections.

*Efficiency:* quantifies the extent to which a network is structurally efficient in exchanging information using shortest paths. To this end, the efficiency between any pair of nodes $(i, j)$ is defined as being inversely proportional to their shortest distance $d_{i, j}$ on the network.

*Global Efficiency (E)* of an unweighted network $\boldsymbol{G}$ with N nodes is defined as the average efficiency in the communication between pairs of nodes, where the average is evaluated over all the couples of nodes $(i, j) \in \boldsymbol{G}$,[85]:

$$E(\boldsymbol{G}) = \frac{1}{N(N-1)} \sum_{i \neq j} \frac{1}{d_{i,j}} = \frac{1}{N} \sum_{i \in G} E_i(\boldsymbol{G}) \tag{6}$$

where in this manuscript we refer to $E_i(\boldsymbol{G})$ as the global efficiency of a node, also known as harmonic closeness centrality,[7].

*Local Efficiency (E_{loc}):* characterizes the local properties of a graph $\boldsymbol{G}$. It is obtained by evaluating the efficiency of each local subgraph $\boldsymbol{G_i}$ for each of the $i = 1, \dots, N$ nodes of the whole graph. Given a node $i$, the subrgraph $\boldsymbol{G_i}$ is constructed by considering the subgraph induced by node $i$ and its $k_i$ first neighbors and by removing node $i$ from this subgraph. Hence, the local efficiency of the network is defined as the average across all the subgraph $Gi$ efficiencies:

$$E_{loc}(\boldsymbol{G}) = \frac{1}{N} \sum_{i \in \boldsymbol{G}} E(\boldsymbol{G_i}) \tag{7}$$

where in this manuscript we refer to $E(\boldsymbol{G_i})$ as the local efficiency of a node.

*Community* is a subset of the graph nodes such that the nodes belonging to the same community are on average more connected than the nodes belonging to different communities. We call community partition a representation of the graph as non-overlapping communities[86].

*Modularity* measures the difference between the fraction of links connecting nodes belonging to the same community in the actual graph and its expected value in a random graph[87]. Hence, the higher is the modularity the more significant is the community partition. For the hyperbrain networks, a natural community partition is defined by separating the nodes in two communities, namely the nodes of P1 (leader) and P2 (follower). In the case of directed weighted graphs with weights $w_{i, j}$, modularity is defined as[87]:

$$Q = \frac{1}{W} \sum_{i,j} \left( w_{i,j} - \frac{s_i^{out} \cdot s_j^{in}}{W} \right) \cdot \delta(C_i, C_j), \tag{8}$$

where $W = \sum_{i,j} w_{i,j}$ (sum of weights), $s_i^{out} = \sum_j a_{i,j} w_{i,j}$ (sum of outward weights), $s_j^{in} = \sum_i a_{i,j} w_{i,j}$ (sum of inward weights), $a_{i, j}$ denotes the $(i, j)$ adjacent matrix elements (binary) and $\delta(C_i, C_j)$ is equal to 1 if $i$ and $j$ belong to the same set of nodes (P1 or P2), and is 0 otherwise.

*Assortativity* measures the tendency of the nodes in a network to be connected to other nodes following similar patterns. In general, if a population of nodes can be divided in different discrete types of nodes according to some countable nodes

characteristic, the assortativity coefficient $r$ is defined as[88]:

$$r = \frac{\sum_i e_{ii} - \sum_i a_i b_i}{1 - \sum_i a_i b_i}, \tag{9}$$

where $e_{ij}$ is the fraction of edges from nodes of type $i$ to nodes of type $j$ (i.e. $\sum_{ij} e_{ij} = 1$), $a_i = \sum_j e_{ij}$ and $b_i = \sum_j e_{ji}$. The assortativity coefficient ranges from −1 to 1. For $r = 1$ we have perfect assortativity, for $r = 0$ the network is non-assortative, while for $-1 \leq r < 0$ we have perfect disassortativity. In particular, we focused on the assortativity of degree, i.e. when $i$ and $j$ are the node degrees.

*Rich-club coefficient:* The rich-club phenomenon in a network describes the tendency of the nodes with a large number of edges (the hubs or rich nodes) to be well-connected to each other, forming tightly interconnected subgraphs (clubs)[89,90]. It can be quantified by computing the so-called rich-club coefficient as a function of the degree $k$ as:

$$\varphi(k) = \frac{2E_{>k}}{N_{>k}(N_{>k} - 1)}, \tag{10}$$

where $N_{>k}$ is the number of nodes in the graph $\boldsymbol{G}$ with degree greater than $k$ and $E_{>k}$ is the number of edges connecting pairs of nodes having degree larger than $k$ (i.e. $\varphi(k)$ is the fraction of such edges actually present in the network, versus the maximum possible number).

*Normalized Rich-club coefficient:* Since nodes with high degrees have a high number of incident edges, they naturally tend to be more densely connected than small degree ones. This effect can be taken into account by defining a normalized version of the rich-club coefficient. As a normalization factor we use the rich-club coefficient $\varphi ran$ $(k)$ computed for the maximal random network obtained through two ends swapping of two edges selected uniformly at random in the original network. Hence, the normalized rich-club coefficient is defined as[90]:

$$\rho_{ran}(k) = \frac{\varphi(k)}{\varphi_{ran}(k)}. \tag{11}$$

If $\rho_{ran}(k) > 1$ for large values of $k$, then starting from a certain $k$, a rich-club phenomenon is present in the network.

**Statistics and reproducibility.** To improve the reproducibility of our findings, we adopted a two-step approach. First, we applied a data-driven analysis method robust to the multiple comparison problem (non-parametric cluster permutation) to identify a frequency-band of interest mostly affected by eye-contact. We conducted this analysis for both inter- and intra-brain connections. Second, we obtained the z-scored matrices of the frequency band revealed by this cluster analysis, which was then used in the network analysis. The network analysis indices were used to compare the network characteristics for both undirected (ciPLV) and directed (PSI) data between friends vs. strangers and leaders vs. followers, which were not used in the first step of the analysis, avoiding circularity. This two-step analysis was used to avoid false positives due to the analysis of multiple network measures on multiple frequency bands.

**Reporting summary.** Further information on research design is available in the Nature Research Reporting Summary linked to this article.

## Data availability
The main data (ciPLV and PSI matrices) (used to generate all the figures and analyses) and the data used for producing Figs. 2c, 4a, b, d, 7a, b are available in OSF (https://osf. io/y9cr2/?view_only=f9d6e2f7549f47dd8076787f499d4252). Individual EEG datasets are available from the corresponding author upon reasonable request.

## Code availability
The EEG data was preprocessed using EEGLAB toolbox (Delorme & Makeig, 2004) in MATLAB (version R2018b). The non-parametric cluster permutation was performed in Matlab using custom written code. The network measures were calculated using Python (version 3.9.5 with Jupyter Notebook 6.4.0. This code is available in OSF (https://osf.io/ y9cr2/?view_only=f9d6e2f7549f47dd8076787f499d4252). Other sections of the code (including code used to collect data) are available from the corresponding author upon reasonable request.

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

## Acknowledgements

This research was funded by BIAL Foundation (No. 138/18). V.L. acknowledges support from the Leverhulme Trust Research Fellowship 278 "CREATE: the network components of creativity and success". We would like to thank Tatiana Adamczewska for her drawing of the paradigm (Fig. 1) and her contribution to data collection. We would like to thank Dr Frederike Beyer for assistance in setting up the study. We would like to thank the students who contributed to data collection in this project, including: Laelle Disu, Lovejot Kaur, and Angeliki Plessa.

## Author contributions

C.D.B.L.: formulated the research question, contributed to the study design, programmed and coordinated data collection, contributed to the analytic ideas, analysed the data, wrote the paper. I.Z.: contributed to the study design, supervised and coordinated data collection, pre-processed the neuroimaging data, contributed to data analysis scripts, contributed to writing the paper. A.G.: contributed to data analysis both conceptually and on producing the analysis scripts, contributed to writing and editing the paper. G.D.B.: contributed to data analysis, conducted graph theoretical analysis of the data, contributed to writing and editing the paper. N.B.: developed the eye-tracking paradigm (with I.M.), contributed in the design of the study; programmed the initial eye-tracking task (modified by C.D.B.L.), contributed to writing and editing the paper. A.C.: contributed to data analysis, conducted graph theoretical analysis of the data, contributed to writing and editing the paper. V.L.: guided the network analysis, contributed to writing and editing the paper. I.M.: developed the eye-tracking paradigm (with N.B.), contributed to the study design and methodological approach, contributed to writing and editing the paper.

## Competing interests

The authors declare no competing interests.
