## [Peer Review File · Communications Biology]

Reviewers' comments:

Reviewer #1 (Remarks to the Author):

The manuscript entitled „Social synchronisation of brain activity by eye-contact authored by Luft, Zion, Giannopoulos, Di Bona, Binetti, Civilini, Latora and Mareschal reports on an investigation of inter- and intra-brain coupling during eye-contact using EEG hyperscanning. Coupling was investigated with simultaneously recorded EEG from two persons who interacted in an eye-contact time estimation task. Results reveal differences in inter- and intra-brain coherence in the gamma-band between an eye-contact condition and a control condition without eye-contact.

The experiment investigates an interesting and timely question of inter-brain coupling of two socially interacting persons and provides potentially interesting results. The manuscript is written well and the experimental setup and task are methodologically sound. Some methodological issues, however, need further clarification and potentially do not hold rigorous evaluation. Additionally, the introduction and discussion are dominated by hyperscanning studies using fMRI and fNIRS, while the methods used in the current experiment are EEG and coherence analysis. Taken together, the manuscript needs thorough revision of the applied methods as well as the introduction and discussion.

Introduction and discussion

Issue 1: References on hyperscanning in the introduction are dominated by studies using fMRI and fNIRS. Though these are valid methods, which allow analysis of inter-brain coupling, they are not used in the current study. EEG and coherence analysis are used in the current study, however the introduction lacks of several important papers using these methods to investigate inter-brain synchrony and of implications drawn from previous EEG hyperscanning studies for the current experiment.

Methods and Results

Issue 2: Intra-brain coherence can be confounded by the effect of volume conduction (e.g. Nolte et al., 2004, Haufe et al., 2013, Bastos & Schoffelen, 2016, Front Syst Neuro). Especially when coherence is computed on the sensor level (between electrodes) the effect of volume conduction can play a major role in driving coherence values. Several approaches were introduced to avoid this problem, which rely e.g. on the non-zero phase-lag part of coherence and can therefore not be driven by volume conduction, which has zero phase-lag, but rather reflects neuronal interactions. Imaginary coherence (Nolte et al., 2004) provides such an approach and should be used to check, which parts of the intra-brain synchronisation are driven by volume conduction. Figure 3 clearly shows neighbouring sensors over the right hemisphere with very high values of coherence, which might be affected. As per definition volume conduction doesn't play a role for coherence between two brains, which are not electrically connected. Therefore, a comparison of coherence values derived from intra- vs. inter-brain synchronisation is not valid.

The analysis of the phase-slope index (PSI) is another example for a method which relies on the imaginary part of coherency and is therefore not affected by volume conduction and mixtures of independent sources (Nolte et al, 2008).

Issue 3: Coherence between neural activity of two persons can be driven by common sensory input, synchronous motor output, or synchronous changes of vegetative physiological signals (cardiovascular activity and breathing). Therefore, the interpretation of synchronous neuronal activity between two brains as reflecting social interaction has to be done with great caution (Hari et al., 2015, Burgess, 2013). A rigorous control of the derived coherence differences is therefore strongly recommended, e.g. compare inter-brain coupling in real interacting dyadic partners with data from randomly shuffled participants.

Issue 4: The synchronisation of the two Starstim EEG amplifiers is not described in the methods. Also the synchronization of triggers should be mentioned. As this is an important issue in EEG hyperscanning, this information should be added.

Reviewer #2 (Remarks to the Author):

Review - Communication Biology

The authors conducted an EEG-based hyperscanning study to investigate the effect of eye-contact. Inter-brain and intra-brain analyses of the undirected and directed synchronization were performed and eye-contact was demonstrated to be associated with inter-brain and intra-brain patterns that were distinct from a control condition and dependent on the relationship between the dyad. Although the results might be interesting, (at least) a substantial revision is necessary, for the following major concerns.

1. While I agree that eye-contact is important for social communication, the link between this specific background and the present experiment is weak. What could be the implication of the experimental paradigm for the research question about eye-contact in social communication? To me, the task for time duration estimation and the corresponding eye fixation task did not resemble the eye contact behavior in our daily life.

2. More explanations on the rationale of the experimental design is needed. For instance, why having the two time duration (1.5 sec vs. 2.5 sec), what was the purpose of having the specific fixations in the control task, etc.

3. Although extensive data analysis was conducted, the necessity of having the inter-brain analysis is not clear (mostly due to the lack of a clear research question).

4. The analysis on the leader-follower further complicated the results. I am confused about the possible social implication of leader-follower in such a simple and abstract task. In other words, what could be the theoretical implications for this piece of result?

Overall, while I appreciate the hard work by the authors, the research question was not clearly presented and therefore it is difficult to evaluate the possible contribution of the present findings.

Reviewer #3 (Remarks to the Author):

In "Social synchronisation of brain activity by eye-contact", Di Bernardi Luft and colleagues present new results about inter-brain synchronization during gaze interaction in humans, combining dual-EEG recording and dual-eye-tracking. The authors studied how friendship and leadership affect both inter- and intra-brain synchronization using two measures — coherence (COH) and phase slope index (PSI)— and network analysis. Their analyses reveal: an increase of gamma band coherence during eye contact compare to the control condition, higher synchronization in friends than strangers, and leader-to-follower inter-brain connectivity in the alpha band. Overall, the paper looks like an interesting addition to the growing literature of hyperscanning during gaze interaction. There are however two major issues to address, and few minor corrections to make.

Major issue #1

The authors used coherence to measure inter-brain synchronization. Such metric mixes amplitude and phase information. Moreover, gamma is known to be easily contaminated by muscle artefacts and saccades, especially in EEG. They thus should demonstrate that the effect reported is not driven by systematic changes in the gamma power. The use of alternative metrics such as Phase Locking Value (PLV) or Circular Correlation (CCOR) should also be considered. Moreover, the gamma band being way above the time-scale of behavior, it seems important that the authors discuss what may be the biological mechanism allowing such tight synchronization between two brains.

Major issue #2

The authors report the effect in the gamma band for COH and in the alpha band for PSI. How do they explain this mismatch? One would indeed expect the same frequency being the signature of information exchange and its direction.

Minor corrections:

- The author must clarify how they handled multiple comparison corrections given the number of graph theory metrics and frequency bands tested.
- "Interbrain" is the anatomical structure corresponding to the posterior division of the forebrain. The hyperscanning community has thus apparently converged in the use of "inter-brain" with a dash.
- In the keywords: "brain sychronization" → "brain synchronization"
- In figures: it is not recommended to use the jet colormap (see: <https://agilescientific.com/blog/2017/12/14/no-more-rainbows>) and bar plots (see: https://barbarplots.github.io/dear_editor.html).

Response to Reviewers

We would like to thank the reviewers for their constructive comments. We addressed each comment in as much depth as possible. We believe that the changes have improved our paper substantially. Notably, in response to a point brought up by two of the reviewers and the editor, we entirely re-analysed all the undirected synchronization data using a measure of phase synchronization which is insensitive to volume conduction: the corrected imaginary part of the phase lag index (ciPLV). We chose this measure due to its robustness to volume conduction, its robustness to noise and its fast speed¹. Please find our point-by-point response below (reviewers' comments in italics and our responses in regular font). Please note that the references in this letter are numbered according to the order they appear in this document (please see reference list at the end), even for references that appear within direct quotes from the manuscript (we did this to facilitate the reading of the response letter).

Reviewers' comments:

Reviewer #1 (Remarks to the Author):

The manuscript entitled „Social synchronisation of brain activity by eye-contact authored by Luft, Zion, Giannopoulos, Di Bona, Binetti, Civilini, Latora and Mareschal reports on an investigation of inter- and intra-brain coupling during eye-contact using EEG hyperscanning. Coupling was investigated with simultaneously recorded EEG from two persons who interacted in an eye-contact time estimation task. Results reveal differences in inter- and intra-brain coherence in the gamma-band between an eye-contact condition and a control condition without eye-contact.

The experiment investigates an interesting and timely question of inter-brain coupling of two socially interacting persons and provides potentially interesting results. The manuscript is written well and the experimental setup and task are methodologically sound. Some methodological issues, however, need further clarification and potentially do not hold rigorous evaluation. Additionally, the introduction and discussion are dominated by hyperscanning studies using fMRI and fNIRS, while the methods used in the current experiment are EEG and coherence analysis. Taken together, the manuscript needs thorough revision of the applied methods as well as the introduction and discussion.

Response: We thank the reviewer for their insightful and constructive feedback on our paper and have addressed their comments in detail below. We believe this has very much improved our paper.

Introduction and discussion

Issue 1: References on hyperscanning in the introduction are dominated by studies using fMRI and fNIRS. Though these are valid methods, which allow analysis of inter-brain coupling, they are not used in the current study. EEG and coherence analysis are used in the current study, however the introduction lacks of several important papers using these

methods to investigate inter-brain synchrony and of implications drawn from previous EEG hyperscanning studies for the current experiment.

Response R1.C1: Thank you for pointing this out. We agree and have amended the introduction throughout to give a clearer perspective on hyperscanning EEG studies. We added some key studies on EEG Hyperscanning and removed unnecessary references to fNIRS and fMRI studies. We only cited fNIRS and fMRI studies which looked at eye-contact. We would like to acknowledge that the Hyperscanning EEG literature is growing fast, so it is possible that we missed relevant papers. If this is the case and the reviewer spots papers we missed please let us know and we will include it/them.

Methods and Results

Issue 2: Intra-brain coherence can be confounded by the effect of volume conduction (e.g. Nolte et al., 2004, Haufe et al., 2013, Bastos & Schoffelen, 2016, Front Syst Neuro). Especially when coherence is computed on the sensor level (between electrodes) the effect of volume conduction can play a major role in driving coherence values. Several approaches were introduced to avoid this problem, which rely e.g. on the non-zero phase-lag part of coherence and can therefore not be driven by volume conduction, which has zero phase-lag, but rather reflects neuronal interactions. Imaginary coherence (Nolte et al., 2004) provides such an approach and should be used to check, which parts of the intra-brain synchronisation are driven by volume conduction. Figure 3 clearly shows neighbouring sensors over the right hemisphere with very high values of coherence, which might be affected. As per definition volume conduction doesn't play a role for coherence between two brains, which are not electrically connected. Therefore, a comparison of coherence values derived from intra- vs. inter-brain synchronisation is not valid.

The analysis of the phase-slope index (PSI) is another example for a method which relies on the imaginary part of coherency and is therefore not affected by volume conduction and mixtures of independent sources (Nolte et al, 2008).

Response R1.C2: We agree with the reviewer regarding the problem of volume conduction in the analysis of intra-brain coherence. That was an oversight on our part especially regarding the comparisons between inter- and intra-brain synchronization. We have reanalysed the data using the corrected imaginary part of the phase lag index (ciPLV). Similarly to the imaginary coherence, this measure is insensitive to zero lag synchronization and is optimised for speed¹. We removed all the previous analyses using coherence and re-wrote the results section using the ciPLV instead. We observed similar findings: an inter-brain and an intra-brain cluster both in the gamma frequency band. The main difference is that since the clusters became stronger, they are also more widespread, with a less defined right parietal topography. Our findings show that gamma synchronization is higher during eye-contact compared to the control condition, especially on the right hemisphere, for both inter- and intra-brain clusters (Figures 2 and 3). Regarding the network analysis, the findings were also similar: significantly higher changes in inter- compared to intra-brain synchronization (degree and network efficiency). We have re-written the entire analysis of

the undirected networks (see Results section) and changed the measure in the methods section.

Issue 3: Coherence between neural activity of two persons can be driven by common sensory input, synchronous motor output, or synchronous changes of vegetative physiological signals (cardiovascular activity and breathing). Therefore, the interpretation of synchronous neuronal activity between two brains as reflecting social interaction has to be done with great caution (Hari et al., 2015, Burgess, 2013). A rigorous control of the derived coherence differences is therefore strongly recommended, e.g. compare inter-brain coupling in real interacting dyadic partners with data from randomly shuffled participants.

Response R1.C3: Thank you for this suggestion, this is a valid point. The problem of the common sensory input is what motivated us to create a closely matched control condition in which the participants still had the same action (and exactly same task parameters) but with different gazing targets. Furthermore, in both conditions we selected the time where the gaze was stable, without moving/changing sensory input. However, this is a good point as it could be that the common sensory input was higher during the eye-contact condition. To address that, we followed your suggestion and created 1000 shuffled datasets for each condition (EC and CTL) where the data of participant 1 of the dyad was matched with participant 2 from another dyad in each condition (as suggested by Burgess, 2013). We hypothesised that if our gamma cluster was caused by a common sensory input, the positive cluster values (EC>CTL) obtained from such shuffled datasets would reflect this difference as phase synchronization would be higher in the shuffled EC compared to CTL. To test this hypothesis, we calculated the cluster statistics for each of these shuffled datasets (the procedures of the cluster permutation analysis were identical to the main analysis but instead of shuffling between labels, we shuffled the pairs as described above – pairing the data of participant 1 in one dyad with participant 2 from another dyad). We tested the significance of our real cluster statistics (our positive eye-contact cluster) against the cluster statistics distribution using the randomly shuffled participants (1000 different datasets). We found that the probability of finding the cluster we observed (real cluster statistic =132.94) using shuffled data was low ($p = .0020$, t-critical = 79.02). Moreover, the t-critical obtained using these shuffled datasets was very similar to the t-critical observed in the main non-parametric cluster permutation (t-critical observed shuffling labels = 76.84). Figure 1 shows the distribution of the cluster statistics obtained using the shuffled data, the t-critical and the real data cluster statistics.

Figure 1. Histogram displaying the distribution of the t-statistics for the cluster analysis under each shuffled dataset. Each datapoint represents the cluster t-statistic values (sum of the cluster t-values) against the cluster using the non-shuffled data (eye-contact cluster in red). The t-critical shows the boundary for statistical significance based on an alpha level of .05.

We have now added this analysis to the paper as an additional control analysis to the results section on page 7:

“To ensure that our effects were not caused by a difference in common sensory input between the conditions (as demonstrated by Burgess²), we created 1000 shuffled datasets for each condition (EC and CTL) where the data of participant 1 of the dyad was matched with participant 2 from another dyad in each condition. We hypothesised that if our gamma cluster was caused by a common sensory input, the positive cluster values (EC>CTL) obtained from such shuffled datasets would reflect this difference as phase synchronization would be higher in the shuffled EC compared to CTL. To test this hypothesis, we calculated the cluster statistics for each of these shuffled datasets (the procedures of the cluster permutation analysis were identical to the main analysis but instead of shuffling between labels, we shuffled the pairs as described above – pairing the data of participant 1 in one dyad with participant 2 from another dyad). We tested the significance of our real cluster statistics (our positive eye-contact cluster) against the cluster statistics distribution using the randomly shuffled participants (1000 different datasets). We found that the probability of finding the cluster we observed (real *cluster t-statistic* =132.94) using shuffled data was low ($p = .0020$, $t\text{-critical} = 79.02$).”

Issue 4: The synchronisation of the two Starstim EEG amplifiers is not described in the methods. Also the synchronization of triggers should be mentioned. As this is an important issue in EEG hyperscanning, this information should be added.

Response R1.C4: Thank you for spotting this gap, we have now added this information to the methods under the subheading “Hyperscanning Setup” (p.21):

“The hyperscanning set up consisted of two connected and synchronized desktops via a crossover

ethernet cable. Each desktop was connected to a different EEG and eye-tracking device. All processes were centralised in Matlab, which was used to stream the eye-tracking data in real time and send triggers to the EEG (via MATNIC toolkit using LSL) based on both the eye-contact task and the eye-tracking behaviour (e.g. saccades detection). During the experimental tasks, both computers streamed data from the eye-tracking machines in real time via Matlab, sending triggers for every event of interest, including saccade detection. The triggers were sent to both EEG machines simultaneously via TCP/IP communication. To test timing precision, we compared the timings (time stamps and samples) when each trigger was received by the EEG machines and the number of samples between two events (e.g. eye-movement/saccade markers) in each of the two files recorded simultaneously. We found no discrepancy, showing that the timings were accurate and that the machines were fully synchronized.”

Reviewer #2 (Remarks to the Author):

The authors conducted an EEG-based hyperscanning study to investigate the effect of eye-contact. Inter-brain and intra-brain analyses of the undirected and directed synchronization were performed and eye-contact was demonstrated to be associated with inter-brain and intra-brain patterns that were distinct from a control condition and dependent on the relationship between the dyad. Although the results might be interesting, (at least) a substantial revision is necessary, for the following major concerns.

Response: Thank you for your insightful review of our paper. We addressed your major concerns which we believe have made our paper stronger, especially in terms of explaining our rationale and outlining its merits and limitations.

1. While I agree that eye-contact is important for social communication, the link between this specific background and the present experiment is weak. What could be the implication of the experimental paradigm for the research question about eye-contact in social communication? To me, the task for time duration estimation and the corresponding eye fixation task did not resemble the eye contact behavior in our daily life.

Response R2.C1: Whilst we agree that our eye-fixation task does not closely resemble the eye-contact behaviour in our daily lives, our study was designed to experimentally isolate (in the best way we could), the eye-contact component of the social interaction, even if this came at the expense of ecological validity. Usually, eye-contact happens in a conversation context or in general social settings. In order to understand the role of specific brain components in eye contact, there is a balance between ecological validity and experimental control. Even though the eye-contact aspect of the study differs from how it occurs in the real world during spontaneous social interactions, our experimental task enabled us to focus on the eye-contact component without confounding it with other aspects of the interaction, which is particularly important when comparing hyperbrain synchronization between friends and strangers. In a behavioural pilot study of this task conducted by two of our co-authors (NB & IM), the participants noticed the social relevance of eye-contact in the task. Our study’s main

contribution is that we were able to isolate the influence of eye-contact on its own while keeping it live (synchronous) and controlling for its duration. The time reproduction task was needed since in real-life we do not sit and stare at someone's eyes. We sought to mimic a non-verbal social interaction punctuated by brief periods of (controlled) eye-contact. To address this important point in the paper, we added the following rationale to the introduction (p.4):

“Most hyperscanning studies report inter-brain synchronization during social interactions when people are face-to-face. Therefore, it is important to investigate the role of eye-contact - a distinguishing feature of face-to-face interactions - in the hyperbrain dynamics, notably focusing on inter- and intra-brain synchronization. Here, we designed an experimental task (Fig.1) which enabled us to isolate the role of eye-contact in order to answer the following research questions: RQ1 “How does eye-contact affect inter- and intra-brain synchronization? RQ2 What are the network characteristics during eye-contact? RQ3 What is the functional significance of these networks? For instance, how do inter- and intra-brain synchronization during eye-contact differ between friends and strangers? RQ4 Is the inter- and intra-brain synchronization during eye-contact directed according to spontaneous leadership roles that emerge in the task?”

We designed an experimental task to isolate the eye-contact from the other elements of the social interaction (Fig 1B), by having participants make a duration reproduction task at the same time. While this task reduces the ecological validity, it was necessary because people do not naturally make uninterrupted eye-contact without doing something else (e.g talking). By giving them a time reproduction task, we tried to minimise the awkwardness of the eye-contact task whilst keeping some elements as close as possible to the characteristics of eye-contact in real life. The time reproduction task also enabled us to measure inter-brain synchronization (EEG) during short time interval bouts of eye-contact similar in duration to those people usually engage in during a face-to-face interaction³. Considering individual differences in relation to preferred eye-contact duration, we chose time intervals of 1.5s and 2.5s which are within the durations found to be comfortable for people⁴. The use of two different durations also enabled us to test whether the participants were truly engaging with the task and whether they changed their estimations based on their partners (e.g. engaged in a leader/follower dynamics). The high time resolution of EEG enabled us to measure phase synchronization during these short bouts of eye-contact. Our behavioural results (*Supplementary S1*) showed that people engaged with the time reproduction task, reproducing lower durations following short intervals and longer durations following longer intervals (Fig.S1A). Our behavioural results showed that: 1) participants underestimated durations during mutual eye-contact compared to the control condition; and 2) during eye-contact, the estimated durations changed according to their partner estimations, which was evidenced by a correlation between the pair's estimations (Fig.S1B). Furthermore, we found that in some pairs, one participant consistently gazed down first during the eye-contact condition, while the other participant followed, and this was not the result of one person simply responding earlier than the other (*Supplementary S2*). We considered the participant who gazed first to be the leader and the one who followed to be the follower, roles that were then used to investigate directed connectivity. The association between gaze following behaviour and leadership has been observed in both non-human⁵ and human animals⁶, and in this study, we investigated whether the direction of the synchrony between brains changes according to people's leader/follower roles.”

We also added a paragraph to our discussion considering the differences between eye-contact in real life and in our task. We considered how these differences could affect our findings to the limitations of study. The text on page 18 reads:

“Despite our attempts to include the key characteristics of eye-contact in social interactions in our task design, replicating its short durations and including a secondary task, our study has limited ecological validity. In naturalistic settings, eye-contact often emerges spontaneously⁷ and it is held for the duration that both people are comfortable doing⁴, whereas in our study, eye-contact was established on demand as a part of the experimental task. A study⁷ observed that during conversation eye-contact is initiated as pupillary synchrony peaks and it

predicts immediate subsequent decline as it breaks. Although our findings do not enable us to speculate about the precursors of establishing or terminating eye-contact, they could guide future analysis looking at the time course or inter-brain of eye-contact in naturalistic settings, especially regarding how undirected gamma and directed alpha networks change as eye-contact initiates, evolves, and terminates. It is of interest to find how these networks relate to changes in pupillary synchrony in naturalistic settings and how each of these components might be associated to Autism Spectrum Disorders (ASD).”

2. More explanations on the rationale of the experimental design is needed. For instance, why having the two time duration (1.5 sec vs. 2.5 sec), what was the purpose of having the specific fixations in the control task, etc.

Response R2.C2: Thank you for pointing this out. We clarified the rationale of our study by adding the paragraphs (listed above). Regarding the two fixation durations, the reason was twofold: 1) We wanted to have a way of checking that the participants were truly engaging with the task. To do that, we needed a clear distinction between two discernible intervals which are not too far apart (but far enough to be distinguishable); 2) Having two different intervals was also important for allowing the emergence of social coordination between participants as observed in the behavioural analysis (Supplementary S1 and S2). We have now added this to our introduction (see above) and also methods (p.20 at the end of *Procedure*):

“The order of the tasks (eye-contact and control) was counterbalanced across pairs. We adopted these two different durations in order to: 1) ensure that the participants engaged with the time reproduction task by comparing how they performed for each duration; 2) analyse a possible interference in behaviour manifested as a change in estimation depending on their partner’s duration. For instance, if one partner hears a short tone, would their estimations be longer if their partner heard a longer duration?”

3. Although extensive data analysis was conducted, the necessity of having the inter-brain analysis is not clear (mostly due to the lack of a clear research question).

Response R2.C3: Our apologies for not having clearly presented our research questions. These are now explicitly stated in the introduction. The analysis of the inter-brain synchronization was central to our study; we wanted to understand and map the hyperbrain networks during eye-contact, and so we needed to examine both the inter- and intra-brain dynamics. One of our key findings is that eye-contact affects the synchronization between two brains more than within each brain, which suggests that eye-contact, even when detached from a naturalistic interaction, is inherently a social signal. Friendship and leadership were associated with differences in inter-brain synchronization rather than intra. We also show that the differences in the network for eye-contact in the inter- and intra-brain connections happened in the same frequency band, which suggests that they might constitute a single network. We believe those findings are relevant for understanding the inter-brain synchronization processes which happen during eye-contact, reflecting the social element present in most studies on social interaction. This is reflected now in our research questions (explicitly stated thanks to your first comment).

4. *The analysis on the leader-follower further complicated the results. I am confused about the possible social implication of leader-follower in such a simple and abstract task. In other words, what could be the theoretical implications for this piece of result?*

Response R2.C4: Our leadership findings were surprising as the behaviour emerged without any instruction to do so, however this reportedly happens in real-world tasks. For example, in joint cooperation tasks leader/follower behaviours emerge naturally without allocating these roles^{8,9}, and we sought to examine the biological basis for how these types of behaviours emerge. We were also interested in understanding whether synchronization during eye-contact could be directed since previous Hyperscanning studies (showing directed inter-brain connectivity from leader to follower) had clear differences in sensory input and motor action during the time of scanning. In our study, the inter- and intra-brain synchronization was measured during static eye-contact, partially removing such change in sensory input and motor action between the dyad. The leadership behaviour is limited to this task though, so we cannot speculate whether this would be the case for more complex leadership behaviours, which is another limitation of our study. We reflected on this issue and added it to the discussion as follows (p.17):

“Our leadership findings warrant further consideration as they are limited to our experimental task. The leadership behaviour emerged naturally, without any instruction to do so, as it is often the case with the emergence of leader/follower roles in naturalistic settings^{8,9}. Our study provides preliminary evidence that inter-brain synchronization direction might predict the leadership roles: with synchronization flowing from leader to follower. We also show that in the hyperbrain, the leaders seem to have more access to the entire network, which deserves further consideration. It would be interesting to understand whether the direction of this association is limited to the task at hand or whether it generalizes to other tasks. Our study was limited to this specific time-reproduction task, so we cannot speculate on whether directed alpha synchronization during eye-contact would predict leadership roles in other tasks, such as dance or even in conversation.”

Overall, while I appreciate the hard work by the authors, the research question was not clearly presented and therefore it is difficult to evaluate the possible contribution of the present findings.

Response R2: We hope that our changes in the introduction, methods, results, and discussion made the rationale of the study, our research questions, and our contributions clearer.

Reviewer #3 (Remarks to the Author):

In “Social synchronisation of brain activity by eye-contact”, Di Bernardi Luft and colleagues present new results about inter-brain synchronization during gaze interaction in humans, combining dual-EEG recording and dual-eye-tracking. The authors studied how friendship and leadership affect both inter- and intra-brain synchronization using two measures — coherence (COH) and phase slope index (PSI)— and network analysis. Their analyses reveal: an increase of gamma band coherence during eye contact compare to the control condition, higher synchronization in friends than strangers, and leader-to-follower inter-

brain connectivity in the alpha band. Overall, the paper looks like an interesting addition to the growing literature of hyperscanning during gaze interaction. There are however two major issues to address, and few minor corrections to make.

Response: Thank you so much for your constructive feedback and very useful comments on our paper. We outline below how we addressed them.

Major issue #1

The authors used coherence to measure inter-brain synchronization. Such metric mixes amplitude and phase information. Moreover, gamma is known to be easily contaminated by muscle artefacts and saccades, especially in EEG. They thus should demonstrate that the effect reported is not driven by systematic changes in the gamma power. The use of alternative metrics such as Phase Locking Value (PLV) or Circular Correlation (CCOR) should also be considered. Moreover, the gamma band being way above the time-scale of behavior, it seems important that the authors discuss what may be the biological mechanism allowing such tight synchronization between two brains.

Response R3.C1: We agree with the reviewer that coherence was not a good choice of synchronization measure so we reanalysed undirected synchronization using another measure, the corrected imaginary part of the phase locking value (ciPLV). This measure is insensitive to zero lag synchronization, robust to noise, and is optimised for speed¹. We removed all the analyses with the coherence and re-wrote the results section using the ciPLV instead. Interestingly, we observed similar findings: an inter-brain and an intra-brain cluster both in the gamma frequency band. The main difference is that since the clusters became stronger, they are also more widespread, less limited to the right parietal region. Our findings show that gamma synchronization is higher during eye-contact compared to the control condition, especially on the right hemisphere, for both inter- and intra-brain clusters (Figures 2 and 3). Regarding the network analysis, the findings were also similar: significantly higher changes in inter- compared to intra-brain synchronization (degree and network efficiency). We have re-written the entire analysis of the undirected networks (see Results section) and changed the measure in the methods section.

Regarding the interpretation, we agree that the potential biological mechanisms underlying inter-brain synchronization in such high frequency is indeed puzzling. Your comment made us rethink the potential mechanisms underlying our results and we have now added a potential explanation (to be followed up in future studies). We added this to the discussion (end of p.16 to page 17), the text reads:

“Considering that eye-contact leads to synchronized eye movements or mimicking behaviour¹⁰, and that gamma band activity was found to be involved in several social skills such as bonding¹¹, empathy¹², mentalizing¹³, cooperation¹⁴, and prosocial behaviour¹⁵, it is possible that the gamma activity involved in processing social cues resets and aligns once eye-contact is established and participants engage in eye-mimicry. For instance, microsaccades happen within the gamma range (their approximate duration is 30ms¹⁶) and were found to induce bursts of high-frequency (gamma) neuronal activity along the visual pathways^{17,18}. Therefore, it is possible that the eye-mimicry is what enables their fast gamma activity to align in phase, which could facilitate the processing of social cues. This is a hypothesis which deserves further investigation, especially in studies with high sampling frequency eye-tracking. This process of resetting gamma phase based on microsaccades mimicry could account for the greater synchrony between friends, since it is possible that the

frequent exposure helps friends learn each other's patterns of movements or microsaccades which could facilitate mimicry and ultimately lead to a synchronous phase reset of gamma band activity."

Major issue #2

The authors report the effect in the gamma band for COH and in the alpha band for PSI. How do they explain this mismatch? One would indeed expect the same frequency being the signature of information exchange and its direction.

Response R3.C2: This is an interesting point which address in the discussion. There are several studies in this field which found inter-brain synchronization in a range of frequency bands within the same paradigm (e.g. ^{19,20}). There are two main candidate explanations for this. The first is that different bottom-up and top-down processes involved in inter-personal synchronization are manifested in undirected higher frequency and directed lower frequency inter-brain synchronization. We added more comprehensive explanations for this reasoning in the discussion (see below). The second explanation is that different types of information are communicated in different frequencies. During eye-contact, we are extracting a wealth of information about the person we are gazing at, including eye-movements, emotions, aesthetics, etc. and it is possible that communication of eye-movements mediating mimicry is communicated at a different frequency compared to, for example, the reading of the others' emotions/intentions/expressions (in our case that could be the other's intention to gaze down).

This is now included in the discussion (p.16-17). The text reads:

"Previous EEG hyperscanning studies revealed increased synchronization during social interactions in a number of different frequencies, most notably in alpha (for a review on EEG hyperscanning studies, see^{21,22}). For example, Dumas et al.¹⁹ observed increases in phase synchronization between a model and an imitator in alpha, beta and gamma frequencies. Another study²³ found that effective social coordination was associated with an increase in synchronized alpha in the right centroparietal regions between a pair. Other studies^{11,14,15,19,24,25} observed increased inter-brain synchrony in higher frequencies. A difficulty in reconciling these frequency differences is that the activities that the pairs perform vary largely between studies, making it harder to disentangle activity reflecting social coordination and task constraints²⁶, especially considering the issues with shared common input². In our experiment, we evaluated synchronization during fixations (when both participants eyes were still), which reduces the influence of joint actions/movement on brain synchronization. Considering that eye-contact leads to synchronized eye movements or mimicking behaviour¹⁰, and that gamma band activity was found to be involved in several social skills such as bonding¹¹, empathy¹², mentalizing¹³, cooperation¹⁴, and prosocial behaviour¹⁵, it is possible that the gamma activity involved in processing social cues resets and aligns once eye-contact is established and participants engage in eye-mimicry. For instance, microsaccades happen within the gamma range (their approximate duration is 30ms¹⁶) and were found to induce bursts of high-frequency (gamma) neuronal activity along the visual pathways^{17,18}. Therefore, it is possible that the eye-mimicry is what enables their fast gamma activity to align in phase, which could facilitate the processing of social cues. This is a hypothesis which deserves further investigation, especially in studies with simultaneous intracranial recordings and high sampling frequency eye-tracking. This process of resetting gamma phase based on microsaccades mimicry could account for the greater synchrony between friends, since it is possible that the frequent exposure helps friends learn each other's patterns of movements or microsaccades which could facilitate mimicry and ultimately lead to a synchronous phase reset of gamma band activity.

We also found that leader to follower directed synchrony occurred in alpha, which is consistent with the physiological interpretation of lower frequencies enslaving higher ones. Although we did not test cross-frequency coupling between the partners, it is possible that the leader's rhythms enslaved the higher frequency eye-contact processes of the follower. Here we suggest that simultaneous synchronization between participants

reflects an increase in synchronization in higher frequencies while directional interactions will be reflected in synchronization in lower frequencies.

An alternative explanation is that different types of information are communicated in different frequencies. During eye-contact, we are extracting a wealth of information about the person we are gazing at, including eye-movements, emotions, aesthetics, etc. and it is possible that the eye-movements mediating mimicry are encoded in a different frequency compared to, for example, the reading of the others' emotions/intentions/expressions. To answer this question, new studies designed to isolate each of these components are needed."

Minor corrections:

- *The author must clarify how they handled multiple comparison corrections given the number of graph theory metrics and frequency bands tested.*

Response R3.C3: The reviewer raises a valid point regarding the multiple comparisons issue. For our pairwise contrasts (same measure, across different ROIs), we applied Bonferroni corrections for multiple comparisons. However, for the network measures, we did not because of known issues with this method for separate measures²⁷ and to reduce the risk of Type II errors. We adopted an approach which triangulates the data with different analyses (and avoids double dipping): we did not draw substantial conclusions based on a single statistical test but across several measures considering the full pattern. For instance, in relation to the network measures, it was clear that eye-contact was associated with higher inter-brain synchronization in friends (a result which was confirmed by all measures such as network strength, degree, density and efficiency), but not with specific differences in network characteristics such as modularity and assortativity. Furthermore, most p-values were not borderline and are coherent with the main picture: higher effects on inter-brain compared to intra-brain connections but no clear differences in topography for friends and strangers/ and for leaders/followers. We have now added this to the end of discussion (p.18) as a point for consideration. The text reads:

"Collectively, our findings support the hypothesis that eye-contact affects the synchronization between two brains more than it affects the links within each brain. It shows that the brains of friends synchronized more strongly but with similar network characteristics. Nonetheless, it is important to interpret some of these findings with caution. First, we looked at a small range of network measures (all reported) without correcting for multiple comparisons. It is possible that type I errors could occur even though most of our p-values are not borderline. We did that in order to avoid type II errors and other known issues with Bonferroni corrections²⁷. We suggest our results to be interpreted as a pattern which shows that eye-contact is associated with higher inter-brain synchronization in friends (a result which was confirmed by all measures such as network strength, degree, density and efficiency), but not with specific differences in network characteristics such as modularity and assortativity."

As for the correction over frequency bands, we conducted the analysis adding all permutation across all frequency bands to the same distribution. Our findings show that the clusters (both ciPLV and PSI, inter- and intra-brain) remain significant at the level of $p < .05$, demonstrating that the clusters are robust across frequencies.

- “Interbrain” is the anatomical structure corresponding to the posterior division of the forebrain. The hyperscanning community has thus apparently converged in the use of “inter-brain” with a dash.

Response R3.C4: Thank you so much for pointing this out. We have now replaced the term throughout the manuscript.

- In the keywords: “brain synchynchronization” —> “brain synchronization”

Response R3.C5: Fixed, thanks so much!

- In figures: it is not recommended to use the jet colormap (see: <https://agilescientific.com/blog/2017/12/14/no-more-rainbows>) and bar plots (see: https://barbarplots.github.io/dear_editor.html).

Response R3.C6: Thank you very much for the materials. We now changed the colourmap of the undirected connectivity topoplots (PLV) to Parula, since the strength of the effects can be easily observed within a range (from low to high), making it suitable for colour deficient readers. However, we did not change the colormaps for the directed synchronization results. The reason for this is that the jet colours are representing, with stronger colours, two opposite directions of synchronization, one represented as strong blue and another with strong red with a yellow around zero and such information would be lost or very difficult to read using the other maps we had available (anything that can be read in gray scale would become confusing as the midpoint would not have the brightest colour). We understand that this might be an issue for individuals who are colourblind, which we tried to mitigate by adding explanations of the patterns observed in the maps to the figure captions. Furthermore, we believe they can still understand the results because the significant cluster connections are also plotted as arrows (in the figure next to the topomaps). The arrows can help the colour deficient reader to understand the patterns we observed without needing to see specific colours.

Regarding the errobars, these have been replaced with violin plots in figures which had two conditions. For many figures though (that had several bars/conditions), this was not possible as the violin plots became cluttered and did not help visualising the data (boxplots are also very confusing for this – so to make this feasible we would have to substantially increase the number of figures in the paper), so we kept them as the traditional error bars.

References

1. Bruña, R., Maestú, F. & Pereda, E. Phase locking value revisited: teaching new tricks to an old dog. *Journal of neural engineering* **15**, 056011 (2018).
2. Burgess, A. P. On the interpretation of synchronization in EEG hyperscanning studies: a cautionary note. *Frontiers in human neuroscience* **7**, 881 (2013).
3. Rogers, S. L., Speelman, C. P., Guidetti, O. & Longmuir, M. Using dual eye tracking to uncover personal gaze patterns during social interaction. *Scientific reports* **8**, 1–9 (2018).
4. Binetti, N., Harrison, C., Coutrot, A., Johnston, A. & Mareschal, I. Pupil dilation as an index of preferred mutual gaze duration. *Royal Society Open Science* **3**, 160086 (2016).
5. Shepherd, S. V., Deaner, R. O. & Platt, M. L. Social status gates social attention in monkeys. *Current biology* **16**, R119–R120 (2006).
6. Liuzza, M. T. *et al.* Follow my eyes: the gaze of politicians reflexively captures the gaze of ingroup voters. *PloS one* **6**, e25117 (2011).
7. Wohltjen, S. & Wheatley, T. Eye contact marks the rise and fall of shared attention in conversation. *Proc Natl Acad Sci USA* **118**, e2106645118 (2021).
8. Van Vugt, M. Evolutionary origins of leadership and followership. *Personality and Social Psychology Review* **10**, 354–371 (2006).
9. D’Ausilio, A. *et al.* Leadership in orchestra emerges from the causal relationships of movement kinematics. *PLoS one* **7**, e35757 (2012).
10. Koike, T., Sumiya, M., Nakagawa, E., Okazaki, S. & Sadato, N. What makes eye contact special? Neural substrates of on-line mutual eye-gaze: a hyperscanning fMRI study. *eNeuro* ENEURO.0284-18.2019 (2019) doi:10.1523/ENEURO.0284-18.2019.
11. Kinreich, S., Djalovski, A., Kraus, L., Louzoun, Y. & Feldman, R. Brain-to-brain synchrony during naturalistic social interactions. *Scientific reports* **7**, 1–12 (2017).

12. Betti, V., Zappasodi, F., Rossini, P. M., Aglioti, S. M. & Tecchio, F. Synchronous with your feelings: sensorimotor γ band and empathy for pain. *Journal of Neuroscience* **29**, 12384–12392 (2009).
13. Cohen, M. X., David, N., Vogeley, K. & Elger, C. E. Gamma- band activity in the human superior temporal sulcus during mentalizing from nonverbal social cues. *Psychophysiology* **46**, 43–51 (2009).
14. Barraza, P., Pérez, A. & Rodríguez, E. Brain-to-Brain coupling in the gamma-band as a marker of shared intentionality. *Frontiers in Human Neuroscience* **14**, 295 (2020).
15. Mu, Y., Han, S. & Gelfand, M. J. The role of gamma interbrain synchrony in social coordination when humans face territorial threats. *Social Cognitive and Affective Neuroscience* **12**, 1614–1623 (2017).
16. Engbert, R., Mergenthaler, K., Sinn, P. & Pikovsky, A. An integrated model of fixational eye movements and microsaccades. *Proc Natl Acad Sci USA* **108**, E765 (2011).
17. Martinez-Conde, S., Macknik, S. L. & Hubel, D. H. Microsaccadic eye movements and firing of single cells in the striate cortex of macaque monkeys. *Nature neuroscience* **3**, 251–258 (2000).
18. Martinez-Conde, S., Macknik, S. L. & Hubel, D. H. The function of bursts of spikes during visual fixation in the awake primate lateral geniculate nucleus and primary visual cortex. *Proceedings of the National Academy of Sciences* **99**, 13920–13925 (2002).
19. Dumas, G., Nadel, J., Soussignan, R., Martinerie, J. & Garnero, L. Inter-brain synchronization during social interaction. *PloS one* **5**, e12166 (2010).
20. Dikker, S. *et al.* Crowdsourcing neuroscience: inter-brain coupling during face-to-face interactions outside the laboratory. *NeuroImage* **227**, 117436 (2021).
21. Liu, D. *et al.* Interactive brain activity: review and progress on EEG-based hyperscanning in social interactions. *Frontiers in psychology* **9**, 1862 (2018).

22. Czeszumski, A. *et al.* Hyperscanning: a valid method to study neural inter-brain underpinnings of social interaction. *Frontiers in Human Neuroscience* **14**, 39 (2020).
23. Tognoli, E., Lagarde, J., DeGuzman, G. C. & Kelso, J. S. The phi complex as a neuromarker of human social coordination. *Proceedings of the National Academy of Sciences* **104**, 8190–8195 (2007).
24. Levy, J., Goldstein, A. & Feldman, R. Perception of social synchrony induces mother–child gamma coupling in the social brain. *Social cognitive and affective neuroscience* **12**, 1036–1046 (2017).
25. Ahn, S. *et al.* Interbrain phase synchronization during turn- taking verbal interaction—a hyperscanning study using simultaneous EEG/MEG. *Human brain mapping* **39**, 171–188 (2018).
26. Hamilton, A. F. de C. Hyperscanning: Beyond the Hype. *Neuron* **109**, 404–407 (2021).
27. Perneger, T. V. What’s wrong with Bonferroni adjustments. *Bmj* **316**, 1236–1238 (1998).

Reviewers' comments:

Reviewer #1 (Remarks to the Author):

The authors greatly improved the manuscript and addressed almost all issues and concerns.

One point is still remaining, which was addressed by Reviewer #3, and not yet fully answered yet, but which I think is also very important. It is crucial to know whether the effect in ciPLV is driven by power differences in the same frequency. E.g. differences in gamma power between the conditions could drive ciPLV differences in the same frequency. Therefore, it is desirable that gamma power is roughly equal between conditions. Is that the case for the contrasts with significant ciPLV differences? Please report.

Reviewer #2 (Remarks to the Author):

The authors made great efforts in revising their manuscript. All my concerns have been properly addressed and I recommend acceptance in its present form.

Reviewer #3 (Remarks to the Author):

I thank the authors for the substantial revision to their original submission. They have tried as best as they can to address all the points raised. Hence, I now endorse the publication of their manuscript.

Response to Reviewers

We would like to thank the editors and the reviewers for their constructive comments, which we think have improved the paper substantially. We are very pleased to hear that the reviewers are happy with the revisions performed and that there is only one remaining point that we need to address in this revision. We have now conducted a new control analysis on gamma power, which made us confident that the differences in gamma phase synchronization are not driven by differences in gamma power between conditions. Please see the final comment and our response below.

Reviewers' comments:

Reviewer #1 (Remarks to the Author):

The authors greatly improved the manuscript and addressed almost all issues and concerns.

One point is still remaining, which was addressed by Reviewer #3, and not yet fully answered yet, but which I think is also very important. It is crucial to know whether the effect in ciPLV is driven by power differences in the same frequency. E.g. differences in gamma power between the conditions could drive ciPLV differences in the same frequency. Therefore, it is desirable that gamma power is roughly equal between conditions. Is that the case for the contrasts with significant ciPLV differences? Please report.

Response: Thank you so much for doing such a careful and constructive review of our work, we really appreciate it. We agree that it is a good idea to make sure gamma power is not the driver of the effects we observed so we conducted a control analysis (now reported in the Supplementary S8) to address this. We added a sentence in our results section to refer to this analysis (p.8):

“To ensure that the differences in phase synchronization (ciPLV) cannot be explained by differences in gamma power between conditions, we conducted a control analysis in which we compared gamma power (absolute and relative) between conditions in each of the measured channels (using the same data used for the ciPLV). Our results (see Supplementary S8) showed no significant differences in gamma power during eye-contact compared to control.”

We reported the analysis in more detail in the supplementary S8:

“To ensure that our findings were not due to differences in gamma power between the conditions (eye-contact vs. control), we compared gamma power between conditions. We estimated the power spectral density using Welch periodogram. For each participant and each channel, we estimated the power spectral density from 4 to 45 Hz (in steps of 1 Hz) in the same data used in the phase synchronization analysis. We analysed both absolute gamma power (30-45 Hz) and relative gamma power by dividing the sum of gamma power between 30 and 45 Hz by total power (sum of power from 4 to 45 Hz). We compared gamma power between eye-contact and control conditions using a paired t-test in each channel (using all participants included in the ciPLV analysis). Since no significant difference was found in any channel for both absolute (Fig.1A) and relative (Fig.1B) gamma power (all t-values < 2 and $p > .05$), there was no need to conduct a non-parametric cluster permutation

(as we had no cluster observed in the real data). This analysis confirms that the effects we observed in the ciPLV are unlikely to derive from differences in gamma power between conditions.

Figure.S8. Differences in gamma power between eye-contact vs. control. A. Topographical distribution of the differences in absolute gamma power between eye-contact and control expressed as t-values (paired t-tests); B. Same as in A but using relative gamma power as the dependent variable. All t-values were lower than 2 and corresponding p-values higher than .05.

Reviewer #2

The authors made great efforts in revising their manuscript. All my concerns have been properly addressed and I recommend acceptance in its present form.

Response: Thank you so much for your careful and constructive review and recommendation.

Reviewer #3

I thank the authors for the substantial revision to their original submission. They have tried as best as they can to address all the points raised. Hence, I now endorse the publication of their manuscript.

Response: Thank you so much for your careful constructive review and recommendation.